



# Changes in Antarctic surface conditions and potential for ice shelf hydrofracturing from 1850 to 2200

Nicolas C. Jourdain[1], Charles Amory[2,1], Christoph Kittel[1], and Gaël Durand[1]

[1]Univ. Grenoble Alpes/CNRS/IRD/G-INP/INRAE, Institut des Geosciences de l'Environnement, Grenoble, France.
[2]Laboratoire des Sciences du Climat et de l'Environnement, LSCE/IPSL, CEA-CNRS-UVSQ, Université Paris-Saclay, 91191 Gif-sur-Yvette, France

**Correspondence:** Nicolas Jourdain (nicolas.jourdain@univ-grenoble-alpes.fr)

**Abstract.**

A mixed statistical-physical approach is used to emulate the spatio-temporal variability of the Antarctic ice sheet surface mass balance and runoff of a regional climate model. We demonstrate the ability of this simple method to extend existing MAR simulations to other periods, scenarios or climate models, that were not originally processed through the regional climate model. This method is useful to quickly populate ensembles of surface mass balance and runoff which are needed to constrain ice sheet model ensembles.

After correcting the distribution of equilibrium climate sensitivity of 16 climate models, we find a likely contribution of surface mass balance to sea level rise of 0.4 to 2.2 cm from 1900 to 2010, and -3.4 to -0.1 cm from 2100 to 2099 under the SSP1-2.6 scenario, versus -4.4 to -1.4 cm under SSP2-4.5 and -7.8 to -4.0 cm under SSP5-8.5. Based on a more limited and uncorrected ensemble, we find a considerable uncertainty in the contribution to sea level from 2000 to 2200: between -10 and -1 cm in SSP1-2.6 and between -33 and +6 cm in SSP5-8.5.

Based on a runoff criteria in our reconstructions, we identify the emergence of surface conditions prone to hydrofracturing. A majority of ice shelves could remain safe from hydrofracturing under the SSP1-2.6 scenario, but all the Antarctic ice shelves could be prone to hydrofracturing before 2130 under the SSP5-8.5 scenario.

## 1 Introduction

In the 21$^{\text{st}}$ century, increasing precipitation over the Antarctic Ice Sheet is expected to compensate a significant part of the dynamical ice mass loss triggered by ocean warming, which mitigates the Antarctic contribution to sea level rise (e.g., Seroussi et al., 2020; Edwards et al., 2021; Fox-Kemper et al., 2021). However, if air temperatures exceed ∼7.5°C over the 1981–2010 average, the increase in accumulation starts to be overwhelmed by the mass loss through liquid water runoff (Kittel et al., 2021; Coulon et al., 2023). This is explained by increasing surface melting and rainfall, and this 7.5°C threshold may be reached by 2100 in the warmest models and scenarios.

The production of liquid water runoff may also be conducive of ice shelf hydrofracturing: in favorable conditions of ice shelf stress, the weight of liquid water can destabilize a fracture and lead to its unstoppable propagation as long as liquid water keeps filling the fracture (Weertman, 1973; Lai et al., 2020). When occurring on ice shelf parts that buttress the upstream



flow, hydrofracturing may strongly enhance the contribution of upstream glaciers to sea level rise (Fürst et al., 2016; Sun et al., 2020). Gilbert and Kittel (2021) estimated that 34% of Antarctic ice shelf area could be vulnerable to collapse at 4°C of warming above pre-industrial levels. The exact warming level needed to trigger important production of runoff on a given ice shelf depends on the amount of snowfall and on the snow/firn temperature and density (Donat-Magnin et al., 2021; van Wessem et al., 2023).

The evolving conditions at the surface of the Antarctic ice sheet and ice shelves are required to drive ice sheet models, from which the contribution of ice sheets to sea level can estimated. In the Ice Sheet Model Intercomparison Project for the 6th Climate Model Intercomparison Project (CMIP6, Eyring et al., 2016; ISMIP6, Nowicki et al., 2016), these conditions were directly calculated from the CMIP model outputs (Nowicki et al., 2020). Despite progresses in their representation of the Antarctic climate (Dunmire et al., 2022), the CMIP models often have a coarse resolution and include a relatively poor

representation of snow processes over ice sheets, in particular with regard to firn saturation by meltwater and subsequent runoff (Nowicki et al., 2020).

    Regional Climate Models (RCMs) dedicated to polar regions and constrained by CMIP projections offer a good alternative to the direct use of CMIP models for the estimation of surface conditions (e.g., Kuipers Munneke et al., 2014; Kittel et al., 2021). Despite detailed snow physics, a major weakness of RCMs is nonetheless the associated requirement for additional

skills and processing/computing time, which is why RCM outputs were not ready on time for ISMIP6-Antarctica (Nowicki et al., 2020; Seroussi et al., 2020). Because of these difficulties, only a limited number of RCM-based projections are usually produced — when produced — which is generally insufficient to sample the CMIP model diversity. The need for weighting or re-sampling the CMIP ensemble to correct unrealistic Equilibrium Climate Sensitivity (ECS, Hausfather et al., 2022) may also require a relatively large number of RCM projections.

Over the years, Antarctic Ice Sheet modellers have often scaled their best estimates of present-day accumulation to temperature anomalies from the CMIP models (e.g., Gregory and Huybrechts, 2006), while Positive Degree Day models have sometimes been used to derive melt rates (e.g., Rodehacke et al., 2020; Zheng et al., 2023). The later are based on daily air temperatures projected by the CMIP models, and can be calibrated to match RCM projections (Coulon et al., 2023). In this paper, we present and evaluate a novel statistical–physical method that emulates the spatio-temporal evolution of the surface

mass balance (SMB) and runoff in Antarctica (section 2). This method is used to extend an ensemble of RCM simulations to other periods, scenarios and CMIP models, and we discuss implications for sea level rise and hydrofracturing from 1850 to 2200 in a probabilistic framework (section 3).



## 2  Methods

Our approach is based on a limited number of RCM simulations that we use to emulate a larger ensemble of surface mass
balance and runoff estimates. For that, we use a statistical-physical method that is presented and evaluated in this section.

### 2.1  Regional climate model projections

We make use of the MAR-3.11 regional climate model (Gallée and Schayes, 1994; Gallée, 1995; Kittel et al., 2021, 2022),
which parameterises multiple processes relevant for polar environments. In MAR, the atmosphere is coupled to a 30-layer
model representing the first 20 m of snow/firn with refined resolution at the surface. The snow/firn model solves prognostic
equations for temperature, mass, water content, and snow properties (dendricity, sphericity, and grain size). In the presence
of surface melting or rainfall, liquid water percolates downward into the next firn layers with a water retention of 5% of the
porosity in each successive layer. The firn layers are fully permeable until they reach a close-off density of $830 \, \mathrm{kg \, m^{-3}}$. To
account for possible cracks in ice lenses and moulins, the part of available water that is transmitted downward to the next layer
decreases as a linear function of firn density, from 100% transmitted at the close-off density to zero at $900 \, \mathrm{kg \, m^{-3}}$ and beyond.
If liquid water is not able to percolate further down, then it fills the entire porosity space of surface layers, and the excess is
considered as runoff and removed from the snow/firn model (there is no representation of ponds or horizontal routing).

The surface mass balance and melting conditions produced by MAR have been evaluated in comparison to observational
products in Agosta et al. (2019), Datta et al. (2019), Donat-Magnin et al. (2020) and Kittel et al. (2021).

The MAR simulations were run at 35 km resolution, and the outputs were conservatively interpolated onto a 4 km stereo-
graphic grid, following the atmospheric forcing protocol in ISMIP6. Unless specified otherwise, we use these 4 km data, and
all spatial integrations presented in this sudy were calculated by accounting for the stereographic scale factor. The ice mask
and the surface elevation are based on Bedmap2 (Fretwell et al., 2013). The actual grounded ice sheet and ice shelf areas are
12.286 and 1.737 million $\mathrm{km^2}$, respectively.

Our MAR simulations cover 1980–2100 and are driven by two CMIP5 and five CMIP6 models under a number of scenarios,
as listed in Tab. 1. The MAR–IPSL-CM6A-LR projection goes until 2200 following the extended SSP5-8.5 scenario (Shared
Socioeconomic Pathways, Meinshausen et al., 2020). The selection of these specific CMIP models was based both on the
availability of 6-hourly outputs (required to provide MAR boundary conditions) and on the evaluation of their present-day
mean characteristics (Agosta et al., 2019; Barthel et al., 2020; Agosta et al., in preparation).

### 2.2  Statistical-physical extension method

Hereafter, we first describe how the RCM projections of surface mass balance, surface melting and runoff production can be
extended in time, and to other scenarios or CMIP models. Then, we evaluate these extended or constructed projections in
comparison to the actual RCM projections.



**Table 1.** List of CMIP models, their ensemble member number, their Equilibrium Climate Sensitivity (ECS, provided by Meehl et al., 2020), and the scenarios for which we have a MAR simulation driven by this CMIP model. The historical MAR simulations only start in 1980, and the projections go until 2100 unless specified otherwise. The model references are provided in section 3.

| CMIP model | era | member | ECS (°C) | Available MAR simulation |
|---|---|---|---|---|
| ACCESS-1.3 | CMIP5 | r1i1p1 | 3.5 | historical, RCP8.5 |
| NorESM1-M | CMIP5 | r1i1p1 | 2.8 | historical, RCP8.5 |
| CESM2 | CMIP6 | r11i1p1f1 | 5.2 | historical, SSP1-2.6, SSP2-4.5, SSP5-8.5 |
| CNRM-CM6-1 | CMIP6 | r1i1p1f2 | 4.8 | historical, SSP5-8.5 |
| IPSL-CM6A-LR | CMIP6 | r1i1p1f1 | 4.6 | historical, SSP5-8.5 until 2200 |
| MPI-ESM1-2-HR | CMIP6 | r1i1p1f1 | 3.0 | historical, SSP1-2.6, SSP2-4.5, SSP5-8.5 |
| UKESM1-0-LL | CMIP6 | r1i1p1f2 | 5.3 | historical, SSP1-2.6, SSP2-4.5, SSP5-8.5 |

### 2.2.1 Rationale

Precipitation in Antarctica mostly consists of snowfall even in a warmer climate (Kittel et al., 2021; Donat-Magnin et al., 2021).
In first approximation, snowfall in Antarctica (SNF) thus increases with air temperature following the Clausius-Clapeyron law, which can be approximated as:

$$\mathrm{SNF}(T_{\mathrm{ref}} + \Delta T) = \mathrm{SNF}(T_{\mathrm{ref}}) \times e^{a\Delta T} \tag{1}$$

where $T_{\mathrm{ref}}$ is a reference air temperature, $\Delta T$ the air warming, and $a$ is typically 0.072 in polar conditions (Donat-Magnin et al., 2021).

Previous modelling studies also found an empirical exponential relationship between surface melt rate (MLT) and air warming:

$$\mathrm{MLT}(T_{\mathrm{ref}} + \Delta T) = \mathrm{MLT}(T_{\mathrm{ref}}) \times e^{b\Delta T} \tag{2}$$

where $b$ is typically between 0.3 and 0.6 in Antarctica (Trusel et al., 2015; Donat-Magnin et al., 2021).

Then, runoff is produced if the melt rate exceeds what can be stored and refrozen in the ongoing snow/firn accumulation
(Pfeffer et al., 1991; Kuipers Munneke et al., 2014; Donat-Magnin et al., 2021; van Wessem et al., 2023). In the absence of rainfall, this is achieved if:

$$\frac{\mathrm{MLT}}{\mathrm{SNF}} \geq r \tag{3}$$

where $r$ is typically between 0.60 and 0.85 depending on the snow properties (Donat-Magnin et al., 2021). The presence of rainfall makes eq. (3) more complex (see Appendix B of Donat-Magnin et al., 2021), but rainfall is generally negligible
compared to snowfall in present-day conditions and to surface melt rates in much warmer conditions (Donat-Magnin et al., 2021). In our method, we therefore assume that precipitation is entirely made of snow, and we neglect sublimation and drifting snow given their small contribution (Agosta et al., 2019).





In this paper, we use these relationships to extrapolate SMB, melt rates and runoff to warmer or colder surface conditions. We assume that all the quantities at surface air temperature $T_{\text{ref}}$ are perfectly known from the RCM, and we want to estimate
them for a temperature change of $\Delta T$ that is provided by a CMIP model. To do so, we use the following sequence of equations, in which RU is the rate of mass loss through runoff ($< 0$), while surface melt rate is defined positive:

$$
\begin{cases}
\text{SNF}(T_{\text{ref}} + \Delta T) &= (\text{SMB}(T_{\text{ref}}) - \text{RU}(T_{\text{ref}})) \times e^{a\Delta T} \\[2ex]
\text{MLT}(T_{\text{ref}} + \Delta T) &= min\{\text{MLT}(T_{\text{ref}}) \times e^{b\Delta T}, m\} \\[2ex]
\text{RU}(T_{\text{ref}} + \Delta T) &= -max\{0, \text{MLT}(T_{\text{ref}} + \Delta T) - r\,\text{SNF}(T_{\text{ref}} + \Delta T)\} \\[2ex]
\text{SMB}(T_{\text{ref}} + \Delta T) &= \text{SNF}(T_{\text{ref}} + \Delta T) + \text{RU}(T_{\text{ref}} + \Delta T)
\end{cases}
\tag{4}
$$

where $a$, $b$ and $m$, and $r$ are the method parameters. The $m$ parameter is introduced to avoid unrealistically high melt rates.

In this work, we use annual means for all the variables. To extend surface variables to a given local warming or cooling
level, we always start from 20 different years (i.e., different values of $T_{ref}$), then we average the 20 extended values. This is done to better sample natural variability and to generate a constructed variability that is mostly related to the CMIP model temperatures.

### 2.2.2   Parameter calibration

The $a$ and $b$ parameters are obtained through a least-mean-square fitting of an exponential curve for SMB minus runoff on the
one hand and the surface melt rate on the other hand. The fit is done on the original model grid as regridding does not preserve exponential relationships. The fitted dataset includes the 1980–2100 period, and the 20-year reference period is 2041–2060. To remove outliers, we only consider points between the 5[th] and the 95[th] percentiles of the SMB minus runoff distribution, and the points where melt rate is greater than its 75[th] percentile. The calibrated $a$ and $b$ parameters are listed in Tab. 2 for individual models.

The maximum local melt rate ($m$ parameter in eq. 4) is set to the 99.99[th] percentile of 1980-2100 local melt rates ($m = 1.80 \times 10^{-4}$ kg.m$^{-2}$.s$^{-1}$). The value of $r$ is more difficult to calibrate as it depends on the density and temperature of snow and on the threshold used to consider that runoff is produced (Pfeffer et al., 1991; Donat-Magnin et al., 2021), so we will assess values covering the aforementioned range, i.e. 0.5 to 0.9.

### 2.2.3   Reconstruction from an earlier period

We first assess the ability of our reconstruction method to extend a 21[st] century RCM projection beyond 2100 based on air temperatures until 2200 from a CMIP model. We consider MAR driven by IPSL-CM6A-LR–SSP5-8.5 over 2101–2200 as our target, and we try to reproduce it from the same MAR simulation before 2100 and the IPSL-CM6A-LR surface air temperatures





**Table 2.** Fit parameters of the reconstructed accumulation and melt rate (see Equ. 4).

| Model | $a$ | $b$ |
|---|---|---|
| MAR–IPSL-CM6A-LR | 0.069 | 0.330 |
| MAR–UKESM1-0-LL | 0.065 | 0.313 |
| MAR–CNRM-CM6-1 | 0.067 | 0.284 |
| MAR–MPI-ESM1-2-HR | 0.071 | 0.335 |
| MAR–CESM2 | 0.065 | 0.303 |
| MAR–ACCESS1-3 | 0.058 | 0.320 |
| MAR–NorESM1-M | 0.080 | 0.358 |
| Mean | 0.068 | 0.320 |

over 2101–2200. For this subsection, we use the $a$ and $b$ parameters of Tab. 2 related to IPSL-CM6A-LR. Here the 20 reference years are taken from 2081 to 2100.

In the original MAR simulation, the grounded ice sheet SMB increases linearly until ~2100 due to increased snowfall, then decreases when surface melt and resulting runoff become important (black lines in the left panels of Fig. 1). The annual grounded ice sheet SMB decreases below the 1995–2014 climatology after 2175 in this simulation (Fig. 1a). Over the ice shelves, the SMB of the original MAR simulation remains steady until ~2100 then drops due to increased runoff with a mass loss rate exceeding $2000 \, \mathrm{Gt} \, \mathrm{yr}^{-1}$ by 2200 (Fig. 1b,d,f).

Over the grounded ice sheet, our reconstruction method captures the melt intensification in the 22$^{\mathrm{nd}}$ century, but at a lower pace than in the original simulation (Fig. 1c). This underestimation is largely due to the inability of our reconstruction method to initiate melting at locations that never experienced melting over 2081–2100. The consequence is a strong SMB overestimation after the first 25 years of reconstruction (Fig. 1a), although this is still an improvement compared to the original IPSL-CM6A-LR outputs given that the overly simple snow component of this model over ice sheets was too simple to calculate runoff (Boucher et al., 2020).

Over the ice shelves, the reconstruction method is very accurate in the first 50 years (Fig. 1b,d,f). The best results over this first period are obtained for $r = 0.6$. After 2150, melt rates in the original MAR simulation start increasing more linearly with time and our reconstruction overestimates melt rates and therefore runoff and mass loss. This is likely due to feedbacks that are represented in the MAR simulations but not in the IPSL-CM6A-LR model. Indeed, the appearance of bare ice with a lower albedo, as well as changes in the cloud radiative properties, have a strong impact on the projected SMB over Greenland (Hofer et al., 2020; Mostue et al., 2023), and a similar effect could be found in warmer conditions in Antarctica (Kittel et al., 2022).

From this first evaluation, we conclude that our reconstruction method is suitable for extension of RCM simulations for approximately 25 years over the grounded ice sheet and 50 years over the ice shelves. Extension to colder conditions in the past are expected to be more accurate. In the following, we use $r = 0.6$ given that it best fits the original simulation over ice shelves until 2150.





**Figure 1.** Evaluation of the extension in time from an earlier period: time series of annual SMB, surface melting and runoff anomalies over the grounded ice sheet (left) and ice shelves (right) until 2200 under the SSP5-8.5 scenario. The black lines show the original MAR–IPSL-CM6A-LR simulation, while the colored lines correspond to the extended 2101–2200 period based on the MAR simulation over 2081–2100 and on the IPSL-CM6A-LR air temperatures over 2101–2200. The extensions are shown for several values of $r$ (see eq. 4) and the biases are indicated for both 2101–2150 and 2151–2200. The anomalies are calculated with respect to the 1995–2014 mean. The time series on this plot are filtered through a 5-year running average.



### 2.2.4 Construction from a warmer scenario

We now assess the ability of our method to reconstruct several scenarios from a MAR simulation driven by a warmer scenario. As mentioned previously, our method is not able to create new melting areas, but it is well suited to decrease melt rates in colder conditions. We therefore consider the three MAR simulations for which we have the SSP1-2.6, SSP2-4.5 and SSP5-
8.5 scenarios (MAR–CESM2, MAR–UKESM1-0-LL, MAR–MPI-ESM1-2-HR), and we evaluate the reconstructions of both SSP1-2.6 and SSP2-4.5 from SSP5-8.5. The calculations are done separately for each of the corresponding simulations (based on the parameters in Tab. 2), but we present the results averaged over the three CMIP models in Fig. 2. Similarly as in the previous subsection, each reconstructed year is the average of 20 reconstructions from a reference ranging from 10 years before to 9 years after the reconstructed year. Doing so, the interannual variability of the extended variables is only attributed
to the air temperature variability in the corresponding scenario.

The SSP1-2.6 and SSP2-4.5 reconstructed fields are quite accurate over the 21$^{st}$ century, even if runoff is significantly underestimated in the last decades. Over the ice shelves, mass loss through increased runoff exceeds mass gain through increased snowfall in SSP5-8.5 in the last quarter of the 21$^{st}$ century (Fig. 2b,f). A moderate mass loss only starts in the last decade of the 21$^{st}$ century for SSP2-4.5 while our reconstruction indicate a moderate mass loss due to underestimated runoff (Fig. 2b,f).
From this evaluation, we conclude that our method is suitable for the reconstruction of multiple SSP scenarios based on an existing MAR simulation in a warmer scenario (here SSP5-8.5).

### 2.2.5 Emulation of simulations driven by other CMIP models

Here we assess the ability of our method to emulate MAR simulations driven by other CMIP models. We consider five MAR simulations driven by different CMIP models under either SSP5-8.5 or RCP8.5, which have a similar radiative forcing, and
we assess the corresponding emulation from the same simulation (for verification of our method) and from six other MAR simulations. We use a similar methodology as in the previous subsection, calculating the average reconstruction of every year from 20 reference years, but instead of using the actual temperature as in the two previous subsections, we use the temperature anomaly with respect to each model's climatology. This was needed given that typical values of surface air temperature may vary from one CMIP model to another, in particular due to differences in the first level height and in their ability to represent
stable surface boundary layers over ice sheets.

First of all, it is verified that biases are small for MAR simulations derived from themselves (Fig. 3), showing that our methodology and its implementation are robust. We nonetheless note significant biases in melt rates when a MAR simulation is derived from another one, exceeding 100% in some cases (Fig. 3c,d), which alters the emulated SMB (Fig. 3a,b). MAR–ACCESS1.3 is an outlier and leads to melt reconstruction with largest biases for the four other models. The realistic SMB
reconstructions derived from MAR–ACCESS1.3 are mostly compensations between overestimated melt and ooverestimated accumulation. The other CMIP5 simulation, MAR–NorESM1-M, is closer to the CMIP6 simulations even if the SMB is generally overestimated over the grounded ice sheet. The CMIP5 models have a relatively low ECS, and starting from MAR





**Figure 2.** Evaluation of the reconstruction from a warmer scenario: time series of annual SMB, surface melting and runoff anomalies over the grounded ice sheet (left) and ice shelves (right) for three scenarios until 2100. The black and grey lines show the original MAR simulations, while the colored lines correspond to the reconstructed SSP1-2.6 and SSP2-4.5 scenarios based on the SSP5-8.5 MAR simulation and the CMIP surface air temperatures. Every line on this plot is the average of three simulations or reconstructions: MAR–CESM2, MAR–UKESM1-0-LL, MAR–MPI-ESM1-2-HR. The anomalies are calculated with respect to the 1995–2014 mean. The time series are filtered through a 5-year running average. The mean and biases over 2015-2100 are indicated on each panel.



forced by any of these models does not give good reconstructions of models like CESM2 or CNRM-CM6-1 that both experience particularly strong warming over the 21$^{st}$ century (Kittel et al., 2021).

Given the biases of the emulation from a single model, we now assess the average of five emulation from different MAR simulations (excluding the simulation itself and the MAR–ACCESS1.3 outlier). The emulation is made for three SSP scenarios, based on the five MAR simulations under the same scenario if available or SSP5-8.5 otherwise. For clarity, we present the average results for MAR–CESM2, MAR–UKESM1-0-LL, MAR–MPI-ESM1-2-HR, which are the three MAR simulations for which we have all three scenarios (Fig. 4).

Averaging the reconstruction from several MAR simulations clearly improves the results, and the reconstructed SMB is generally quite accurate throughout the 21$^{st}$ century (Fig. 4a,b), although the mass loss at the surface of ice shelves is overestimated under SSP5-8.5 due to overestimated melt and runoff (Fig. 4d,f). From this evaluation, we conclude that our multi-model emulation method is suitable for the emulation of MAR simulations driven by CMIP models that have never actually been used to drive MAR simulations. It is important to stress that this method only gives meaningful results because of the average over

several GCMs, possibly due to the various responses of clouds and snow albedo from one model to another for a given warming level (Kittel et al., 2022).







**Figure 3.** Evaluation of the reconstruction method from other models for SMB (upper panels) and surface melting (lower panels). The five radial lines correspond to five MAR simulations driven by different CMIP6 or CMIP5 models (thick black pentagons), which are emulated from other available MAR simulations and the corresponding CMIP model surface temperatures (colored pentagons). The thin grey pentagons indicate the SMB or melt rate iso-value. The anomalies are calculated over 2015–2100 with respect to 1995–2014 under the SSP5-8.5 or RCP8.5 scenarios.




**Figure 4.** Evaluation of the reconstruction method from five other models: time series of annual SMB, surface melting and runoff anomalies over the grounded ice sheet (left) and ice shelves (right) for three scenarios until 2100. The black and grey lines show the original MAR simulations for three SSP scenarios, while the colored lines correspond to the reconstruction of these simulations from 5 independent models. Every line on this plot is the average of three simulations or reconstructions: MAR–CESM2, MAR–UKESM1-0-LL, MAR–MPI-ESM1-2-HR. The anomalies are calculated with respect to the 1995–2014 mean. The time series are filtered through a 5-year running average. The mean and biases over 2015-2100 are indicated on each panel.





## 3 Ensemble projections from 1850 to 2200

### 3.1 Approach

Our simulations over 1980–2100 are based on MAR simulations or emulations driven by 16 CMIP6 models listed in Tab. 3. We
also reconstruct the historical period from 1850 to 1979 for these models based on the 1981–2000 period which is the earliest
period covered by the MAR simulations (method described in section 2.2.3).

For the projections between 2101 and 2200, we have a single MAR simulation forced by IPSL-CM6A-LR under SSP5-
8.5 due to the non availability of 6-hourly 3-dimensional atmospheric data beyond 2100 for most CMIP models. The CMIP
surface air temperatures are available until 2300 for seven other simulations for the SSP1-2.6 and SSP-5.85 pathways (see stars
in Tab. 3). For these seven simulations, we first extend the simulations of Tab. 3 from the 2081–2100 (Tab. 3) to 2101–2120
using the method described in section 2.2.3. Then we emulate the $22^{nd}$ century based on the CMIP model temperatures and the
MAR–IPSL-CM6A-LR simulation using the method described in section 2.2.5, and we apply a ramping transition between the
two methods from 2101 to 2120. The MAR–UKESM1-0-LL simulation until 2200 is reconstructed for the ensemble member
r4i1p1f2 (the only one available beyond 2100) from the actual MAR–UKESM1-0-LL r1i1p1f2 simulation until 2100 and from
MAR–IPSL-CM6A-LR after 2100, with a 20-year transition.

The median ECS of this 16-model ensemble is 4.25°C, and the ECS of three models out of 16 exceeds 5°C, which is high
compared to the best estimate of 3.0°C and the 90% confidence interval of 2.0-5.0°C estimated in the IPCC $6^{th}$ Assessment
Report (Forster et al., 2021). To build a realistic ensemble mean, we therefore attribute weights to individual models, which
are the probability of a skew-normal distribution fitted to obtain the $5^{th}$, $50^{th}$ and $95^{th}$ percentiles at ECS of 2.0, 3.0 and 5.0°C
(Fig. 5). The corresponding weights are listed in Tab. 3. Despite the imperfect model sampling, the weighted 16-model mean
ECS falls from 4.0°C to 3.3°C thanks to weighting, which is closer to the best ECS estimates (Forster et al., 2021) and to the
CMIP5 multi-model mean (3.2°C in Meehl et al., 2020). To calculate the percentiles of the weighted distribution, we consider
a number of values equal to 100 times the weight for each model, which shifts the multi-model ECS distribution much closer
to the ECS very likely range (see small triangles in Fig. 5).

It is more difficult to weight the ensemble for the extension to 2200 as only 8 models are available, with a distribution that is
much biased towards high ECS values. Five models indeed have an ECS greater than 4.5°C and the three other models would
take more than 70% of the weight with the previous method without even covering very low ECS. In the following, we will
therefore present the eight runs as a range of possibilities without any weighting.

### 3.2 Grounded ice sheet and sea level

First of all, our reconstructed ensemble indicates that the grounded ice sheet SMB was 110 $Gt\,yr^{-1}$ lower in 1850–1869
than in 1995–2014 and increased slowly through the $20^{th}$ century (Fig. 6a). In comparison, a combination of ice cores and
simulations from another regional climate model gave $\sim$200 $Gt\,yr^{-1}$ of difference between these two periods (Thomas et al.,
2017). This increasing SMB in our reconstruction is equivalent to a reduction of 1.3 cm of global mean sea level from 1900 to





**Table 3.** CMIP6 models used to drive MAR simulations or emulations until 2100 in section 3. The ECS values are from Meehl et al. (2020), except for NorESM2-MM which is from Seland et al. (2020). Stars beside model names indicate that the CMIP6 simulations were extended to 2300 under the SSP1-2.6 and SSP-5.85 pathways. The entries for the three SSP pathways indicate whether it was derived from the actual MAR simulation driven by this CMIP model under this scenario ("MAR"), from a MAR simulation driven by this CMIP model but for a warmer scenario ("from SSP5-8.5", i.e. method described in section 2.2.4), or reconstructed from six MAR simulations driven by different CMIP models ("from 6 models", i.e. method described in section 2.2.5). The historical period was directly available from the five CMIP models for which at least a MAR projection was available, and it was reconstructed from 6 models for the other CMIP models.

| CMIP model | Member | Reference | ECS | weight | SSP1-2.6 | SSP2-4.5 | SSP5-8.5 |
|---|---|---|---|---|---|---|---|
| ACCESS-CM2 ⋆ | r1i1p1f1 | Bi et al. (2020) | 4.7°C | 0.11 | from 6 models | from 6 models | from 6 models |
| ACCESS-ESM1-5 ⋆ | r1i1p1f1 | Ziehn et al. (2020) | 3.9°C | 0.24 | from 6 models | from 6 models | from 6 models |
| CanESM5 ⋆ | r1i1p1f1 | Swart et al. (2019) | 5.6°C | 0.03 | from 6 models | from 6 models | from 6 models |
| CESM2 | r11i1p1f1 | Danabasoglu et al. (2020) | 5.2°C | 0.06 | MAR | MAR | MAR |
| CESM2-WACCM ⋆ | r1i1p1f1 | Gettelman et al. (2019) | 4.8°C | 0.10 | from 6 models | from 6 models | from 6 models |
| CNRM-CM6-1 | r1i1p1f2 | Voldoire et al. (2019) | 4.8°C | 0.10 | from SSP5-8.5 | from SSP5-8.5 | MAR |
| CNRM-ESM2-1 | r1i1p1f2 | Séférian et al. (2019) | 4.8°C | 0.10 | from 6 models | from 6 models | from 6 models |
| GFDL-CM4 | r1i1p1f1 | Held et al. (2019) | 3.9°C | 0.24 | - | from 6 models | from 6 models |
| GFDL-ESM4 | r1i1p1f1 | Dunne et al. (2020) | 2.6°C | 0.47 | from 6 models | from 6 models | from 6 models |
| GISS-E2-1-H ⋆ | r1i1p1f2 | Kelley et al. (2020) | 3.1°C | 0.41 | from 6 models | from 6 models | from 6 models |
| INM-CM5-0 | r1i1p1f1 | Volodin et al. (2017) | 1.9°C | 0.18 | from 6 models | from 6 models | from 6 models |
| IPSL-CM6A-LR ⋆ | r1i1p1f1 | Boucher et al. (2020) | 4.6°C | 0.12 | from SSP5-8.5 | from SSP5-8.5 | MAR |
| MPI-ESM1-2-HR | r1i1p1f1 | Müller et al. (2018) | 3.0°C | 0.43 | MAR | MAR | MAR |
| MRI-ESM2-0 ⋆ | r1i1p1f1 | Yukimoto et al. (2019) | 3.2°C | 0.39 | from 6 models | from 6 models | from 6 models |
| NorESM2-MM | r1i1p1f1 | Seland et al. (2020) | 2.5°C | 0.47 | from 6 models | from 6 models | from 6 models |
| UKESM1-0-LL ⋆ | r1i1p1f2 | Sellar et al. (2020) | 5.3°C | 0.05 | MAR | MAR | MAR |

2010 (likely range: 0.4 to 2.2 cm) with respect to the 1891–1910 mean (i.e., assuming that the climatological SMB over that period contributed to zero sea level rise).

Our projections over the grounded ice until 2100 agree quite well with previous estimates of sea level contribution reported by the IPCC for the three scenarios, and are slightly weaker than previous estimates for the unweighted CMIP6 ensemble (Tab. 4). We estimate a median SMB mitigation of sea level rise between 2.0 and 5.0 cm depending on the scenario. As in Kittel et al. (2021), the three SSP scenario diverge after 2040: the SSP1-2.6 SMB remains ∼100 Gt yr$^{-1}$ above 1995–2014, the SSP2-4.5 SMB keeps increasing until 2100 at a rate similar to 1970–2014, and the SSP5-8.5 SMB increases 1.7 times faster than SSP2-4.5 until 2100 (Fig. 6a).

For seven out of eight models going beyond 2100, the maximum SMB over the grounded ice sheet is reached between 2090 and 2120 under SSP5-8.5 and a few decades earlier under SSP1-2.6 (Fig. 8a). Under SSP1-2.6, the SMB over the grounded ice sheet goes back to present-day values during the 22$^{nd}$ century. Under SSP5-8.5, the three models with an ECS lower than





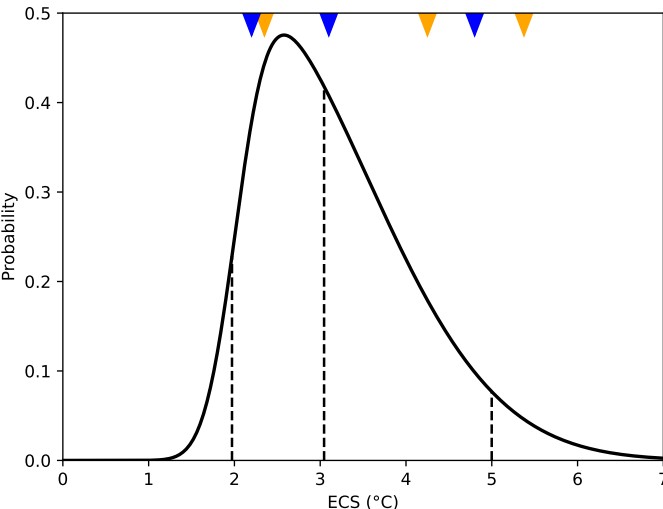

**Figure 5.** Skew-normal ECS probability (solid line) fitted to obtain its $5^{th}$, $50^{th}$ and $95^{th}$ percentiles at 2.0, 3.0 and 5.0°C (dashed lines). The orange triangles indicate the $5^{th}$, $50^{th}$ and $95^{th}$ percentiles of the ECS of the unweighted 16-CMIP model distribution, and the blue triangles show the equivalent for the weighted distribution. The skew-normal distribution was generated using the `skewnorm.pdf` function of the `scipy.stats` package (Virtanen et al., 2020), with a skewness parameter of 5.08, an offset parameter (`loc`) of 2.02°C, and a scale parameter of 1.52.

**Table 4.** Projected sea level contributions (in cm) from the Antarctic Ice Sheet SMB from 2000 to 2099 (relative to 1995–2014, i.e. assuming that the mean SMB over that period yields no sea level rise), for the three selected SSP scenarios, shown as median (likely range, i.e., 17–83th percentile) [very likely range, i.e., 5–95th percentile]. The IPCC-AR5/6 estimates are those presented in Tab. 9.3 of IPCC-AR6 (Fox-Kemper et al., 2021), i.e., recalculated for the SSP scenarios from IPCC-AR5, and originally derived from the CMIP5 global mean surface air temperature using a linear accumulation-temperature relationship (Church et al., 2013). The data of Kittel et al. (2021) are statistical reconstructions based on the air temperature averaged south of 60°S for 33 CMIP6 models, and the percentiles have been recalculated for this table.

| Study | SSP1-2.6 | SSP2-4.5 | SSP5-8.5 |
|---|---|---|---|
| IPCC AR5/6 | –2  (–3 to –1) [-4 to -1] | –3  (–4 to –2) [-6 to -1] | –5  (–7 to –3) [-9 to -2] |
| CMIP6 estimate by Kittel et al. (2021) | -2.6 (-4.0 to -1.6) [-4.8 to -1.1] | -3.9 (-5.2 to -2.5) [-5.9 to -1.8] | -5.7 (-8.1 to -3.8) [-8.7 to -3.2] |
| This study | -2.0 (-3.4 to -0.1) [-4.1 to 0.1] | -3.2 (-4.4 to -1.4) [-6.1 to -0.5] | -5.0 (-7.8 to -4.0) [-8.8 to -2.6] |





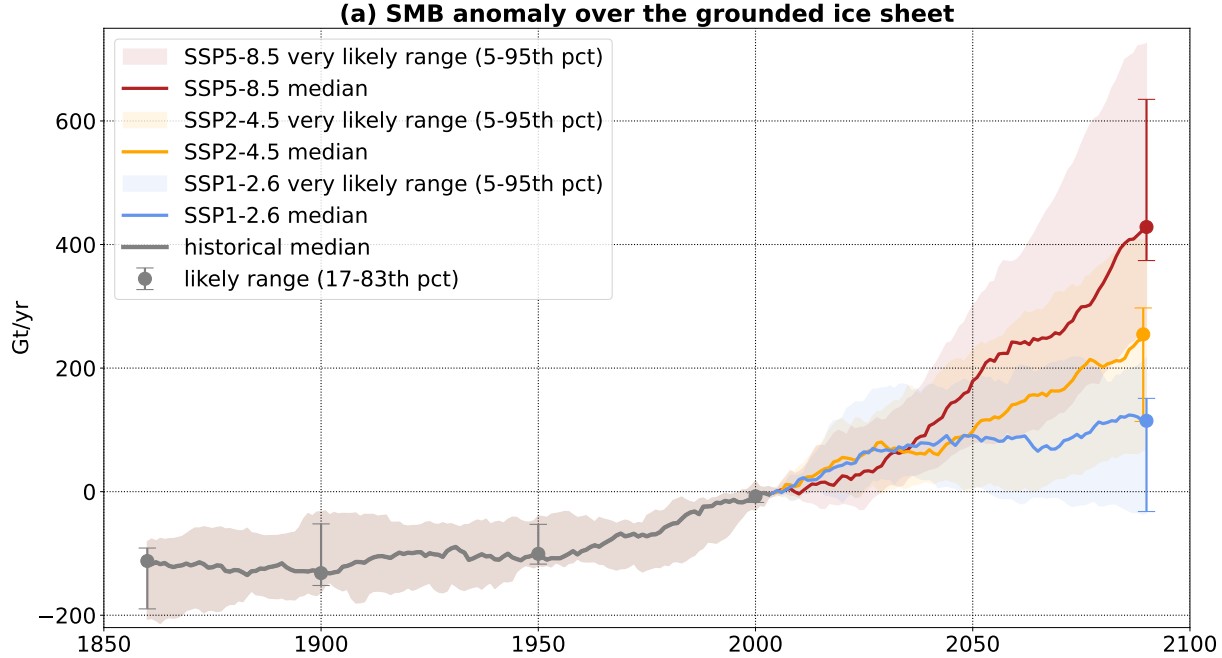

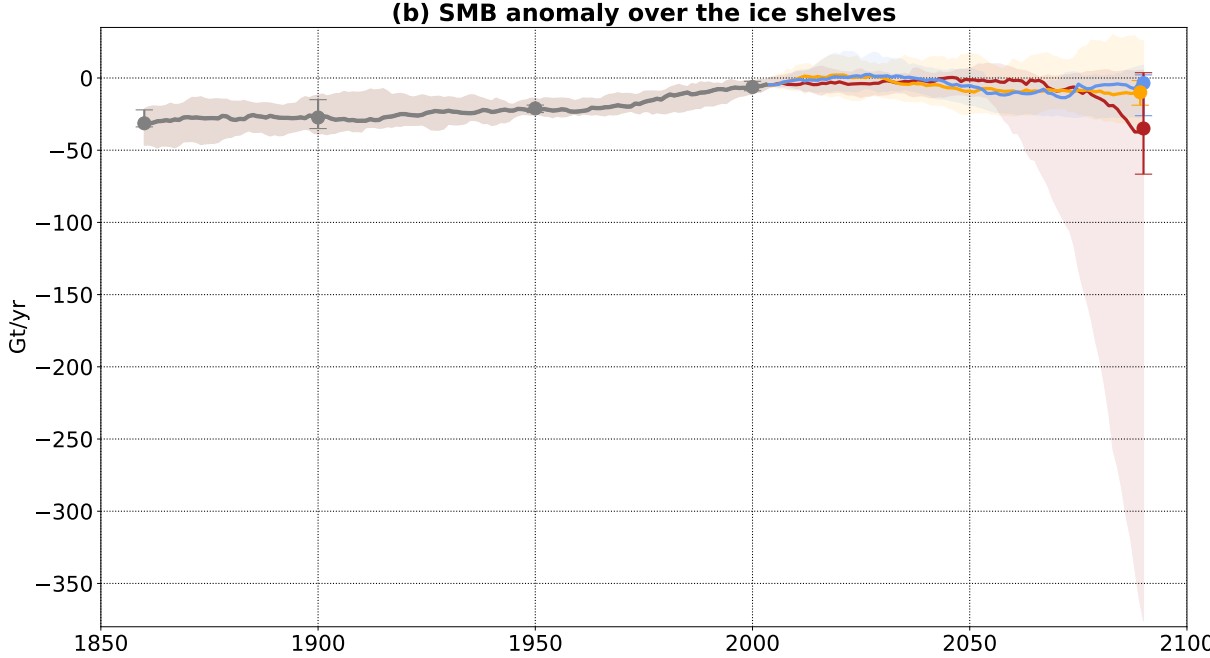

**Figure 6.** Reconstructed ensemble of surface mass balance over the grounded Antarctic ice sheet (a) and ice shelves (b) for the historical period and three SSP scenarios. The median and percentiles are calculated based on the 16-model ensemble weighted to match with the very likely range of ECS (see section 3.1). A 21-year running average has been used for all the time series.





**(a) 1851-1870 SMB anomaly w.r.t. 1995-2014**  **(b) SSP1-2.6 2081-2100 SMB anomaly w.r.t. 1995-2014**

**(c) SSP2-4.5 2081-2100 SMB anomaly w.r.t. 1995-2014**  **(d) SSP5-8.5 2081-2100 SMB anomaly w.r.t. 1995-2014**

**Figure 7.** Weighted mean SMB anomaly for four different periods or scenarios with respect to the average SMB over 1995–2014. This was calculated from the 16 models listed in Tab. 3. The grey contours indicate the topography (every 1000 m) and the black contours show the ice shelves.





**Table 5.** Projected sea level contributions (in cm) from the Antarctic Ice Sheet SMB from 2000 to 2200 (relative to 1995–2014).

| Model | SSP1-2.6 | SSP5-8.5 |
|---|---|---|
| MAR–ACCESS-CM2 | -5.3 | -22.7 |
| MAR–ACCESS-ESM1-5 | 1.1 | -17.5 |
| MAR–CanESM5 | -1.3 | 6.0 |
| MAR–CESM2-WACCM | -10.3 | -0.6 |
| MAR–GISS-E2-1-H | -4.8 | -32.9 |
| MAR–IPSL-CM6A-LR | -4.0 | -16.9 |
| MAR–MRI-ESM2-0 | -3.2 | -22.2 |
| MAR–UKESM1-0-LL | -5.7 | -15.8 |

4°C predict a SMB that remains over the present-day value until 2200. In contrast, four models predict that increasing runoff over the grounded ice sheet will overwhelm increasing accumulation, with a SMB decreasing below the present-day value after 2035 to 2075 depending on the model. It cannot be ruled out that the grounded ice sheet reaches a net surface mass loss near 2200, although this is extremely unlikely given that the two models crossing or approaching this limit are above the 95[th] percentile at the end of the 21[st] century.

In terms of sea level, the net contribution of the Antarctic SMB over 2000–2200 is between -10 and -1 cm for SSP1-2.6 and between -33 and +6 cm for SSP5-8.5 (Tab. 5). Interestingly, the relative importance of sea level reduction between SSP1-2.6 and SSP5-8.5 is reversed for the two models producing the largest amount of runoff in the 22[nd] century (MAR–CanESM5 and MAR–CESM2-WACCM) compared to the other models. This is due to the massive runoff production over the grounded ice sheet after 2150 under SSP5-8.5 in these two models, which counterbalances the excess of accumulation before 2150 (Fig. 8a). 250 MAR–CanESM5 even has a net positive contribution to sea level rise over the two centuries under SSP5-8.5.

In terms of patterns, the projected increase in SMB over the grounded ice sheet until 2100 is largest along the coast of the Bellingshausen and Amundsen seas, as well as in Dröning Maud Land (Fig. 7. The models producing large amounts of runoff by 2200 under SSP5-8.5 tend to have weaker SMB than presently below 1000 m above sea level, i.e. along the coastline and upstream of many ice shelves (Fig. 9.

**3.3 Ice shelves and potential for hydrofracturing**

Our projections indicate that the SMB over ice shelves has slightly increased over the 19[th] and 20[th] centuries, and it is not expected to significantly evolve throughout the 21[st] century under SSP1-2.6 and SSP2-4.5 (Fig. 6b). The SMB even increases by a few tens of $\mathrm{Gt\,yr^{-1}}$ along the 22[nd] century under SSP1-2.6 (Fig. 8b). The SSP5-8.5 SMB projections over ice shelves have a considerable spread, with a median close to present-day values and most of the weighted distribution within the historical range until 2100, but the 5[th] percentiles approaches -400 $\mathrm{Gt\,yr^{-1}}$ of anomaly by 2100 (Fig. 6b). This is due to the emerging 260 importance of ice-shelf runoff at the end of the 21[st] century in the warmest simulations. The ice shelves under SSP5-8.5 experience a net surface mass loss after 2090 to 2125 (depending on the model), to the exception of MAR–GISS-E2-1-H that



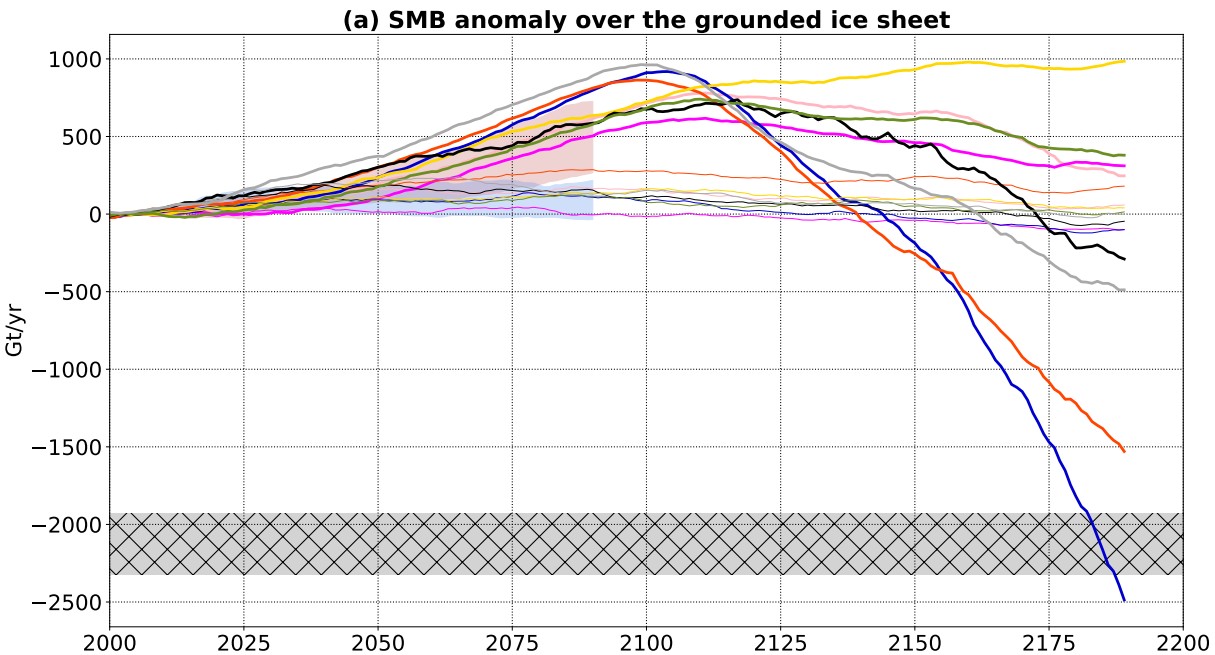

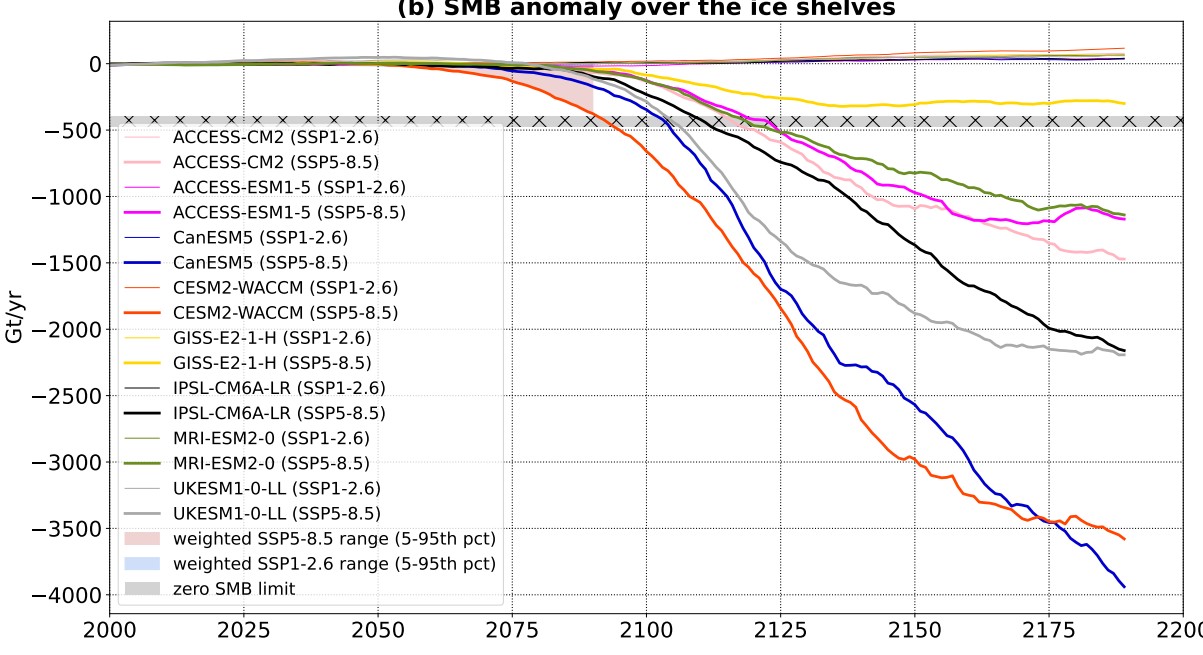

**Figure 8.** Eight reconstructions of surface mass balance over the grounded Antarctic ice sheet (a) and ice shelves (b) for the SSP-1.26 and SSP5-8.5 scenarios. The very likely range from 16 reconstructions over 2000–2100 (same as Fig. **??** is also shown. The hatched area indicates the anomaly interval at which SMB reaches zero, according to the MAR, RACMO and HIRHAM present-day values reported in Mottram et al. (2021). A 21-year running average has been used for all the time series.




**SSP5-8.5 2181-2200 SMB anomaly w.r.t. 1995-2014**
**(a) CanESM5, CESM2-WACCM, IPSL-CM6A-LR, UKESM1-0-LL**

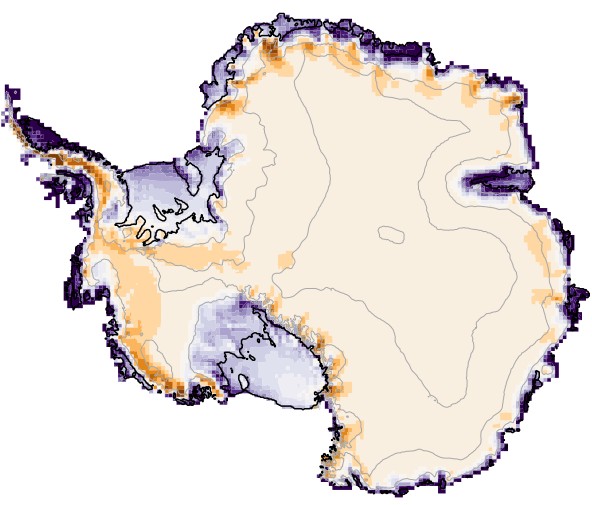

**(b) ACCESS-CM2, ACCESS-ESM1-5, GISS-E2-1-H, MRI-ESM2-0**

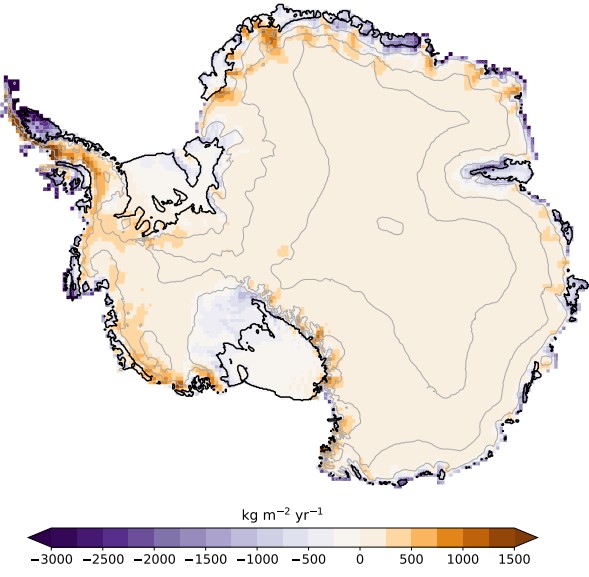

**Figure 9.** Mean 2181–2200 SMB anomaly under SSP5-8.5 with respect to the average SMB over 1995–2014, for (a) the four models with the most negative SMB spatially integrated anomaly and (b) the four models with the least negative spatially integrated SMB anomaly. This was calculated from the 8 models listed in Tab. 3 with equal weight for all models. The grey contours indicate the topography (every 1000 m) and the black contours show the ice shelves.





stabilizes slightly above the zero SMB limit. The most extreme surface mass loss at 2200 is reached by MAR–CanESM5, which has the highest ECS of our ensemble, and is equivalent to an average ice shelf thinning rate of $2 \text{ m yr}^{-1}$ (assuming

an ice density of $920 \text{ kg m}^{-3}$). Spatially, a net surface mass loss arises for several ice shelves of the Antarctic Peninsula by 2100 in the various scenarios (Fig. 7), and net surface mass loss becomes more widespread around Antarctica after 2100 under SSP5-8.5 (Fig. 9). In East Antarctica, net surface mass loss first appears near the grounding line (Fig. 7), likely due to the diabatic heating of downsloping katabatic winds and enhanced melt-albedo feedback, as previously observed by Lenaerts et al. (2017).

We now investigate the emergence of surface conditions that would be necessary to permit hydrofracturing, keeping in mind that mechanical conditions would also be necessary for the developments of fractures (see Introduction). As discussed previously, melting alone is not always sufficient to saturate the firn with liquid water, which is why we do not consider that a necessary condition for hydrofracturing can be based on melt rates. Instead of a melt rate threshold such as the one used in ISMIP6 (Nowicki et al., 2020; Seroussi et al., 2020), we therefore use a runoff threshold.

Little is known about the amount of runoff needed to induce widespread hydrofracturing of an ice shelf, as processes are complex and involve local ice shelf flexure associated with melt water ponding and drainage (Banwell et al., 2013, 2019). Multiple lines of evidence nonetheless suggest that widespread melting over the Larsen B Ice Shelf contributed to its abrupt collapse in 2002 (Rack and Rott, 2004; van den Broeke, 2005; Sergienko and Macayeal, 2005; Robel and Banwell, 2019). The average runoff production over Larsen B prior to its collapse was estimated between 200 and $300 \text{ kg m}^{-2} \text{ yr}^{-1}$ (van Wessem

et al., 2016; Costi et al., 2018), so our runoff threshold has to be lower. We empirically set the threshold to $100 \text{ kg m}^{-2} \text{ yr}^{-1}$ and we will discuss the sensitivity to this choice in the Discussion section. Extreme runoff events, such as those induced by atmospheric rivers, may be sufficient to trigger hydrofracturing (Wille et al., 2022), but our ensemble is too small to build reliable statistics of such events, and we consider runoff production averaged over 10 years. Finally, very warm conditions may lead to runoff production over the grounded ice sheet, as high as ∼1000 m high (Fig. 10), and a large part of this runoff

is expected to drain into ice shelves located downstream (Kingslake et al., 2017). In the absence of a sophisticated surface hydrology model, we consider that all the runoff produced in a drainage basin (such as indicated in Fig. 10) will uniformly cross the coastline of this drainage basin, and the fraction received by the ice shelves of this drainage basin corresponds to the fraction of this coastline occupied by ice shelves. In summary, the necessary condition for hydrofracturing is reached at the end of the first 10-year period for which average runoff exceeds $100 \text{ kg m}^{-2} \text{ yr}^{-1}$, accounting for both the runoff produced over

the entire ice shelf and the runoff flowing from upstream glaciers.

According to our estimates, a few ice shelves were already likely (Fig. 10) or very likely (Fig. 11) in conditions favorable to hydrofracturing before 2015, and sometimes since 1850. This is the case of Larsen A and B that collapsed in 1995 and 2002 (Rott et al., 1996, 2002) after a progressive thinning that lead to favorable mechanical conditions for fractures (Shepherd et al., 2003). This is also the case of the Georges VI Ice Shelf over which melt ponds have been observed since the 1960s

(Wager, 1972; Bell et al., 2018; Banwell et al., 2021) without leading to a widespread collapse although it lost 8% of its area from 1947 to 2010 (Cook and Vaughan, 2010). For the Wilkins Ice Shelf, that progressively thinned (Braun et al., 2009) before a partial disintegration in 2008 likely due to hydrofracturing (Scambos et al., 2009), our likely range starts in 1906 and



has a considerable spread because of present-day values relatively close to the threshold. Such a large likely range extending throughout the 20[th] century is also found for the nearby Wordie Ice Shelf, which disintegrated from 1991 to 2009 (Doake and Vaughan, 1991; Cook and Vaughan, 2010), and Larsen C Ice Shelf, which lost 25-30% of its area from the 1970s to 2021 (Cook and Vaughan, 2010; Greene et al., 2022).

In our projections, conditions favorable to hydrofracturing become likely by ∼2050 for several East Antarctic ice shelves (Fig. 10). This is the case of Nivl, Roi Baudouin, Amery and Shackleton ice shelves on which widespread melt ponds or aquifers are already observed in present-day conditions (Kingslake et al., 2017; Bell et al., 2018; Stokes et al., 2019). Under SSP1-2.6, more than half of the 56 monitored ice shelves are unlikely to experience hydrofracturing until 2200. In contrast, most ice shelves cross the threshold before 2100 under SSP5-8.5. The giant Ross and Ronne-Filchner ice shelves are unlikely to experience hydrofracturing before the early 22[nd] century. In the Amundsen Sea, the ice shelves from Thwaites to Getz are unlikely to experience hydrofracturing before the very end of the 21[st] century, as previously suggested by Donat-Magnin et al. (2021). These spatial patterns in the years of emergence are consistent with those found by Kuipers Munneke et al. (2014), but we find that all the ice shelves are likely to experience conditions favorable to hydrofracturing after 2100 under SSP5-8.5 while Kuipers Munneke et al. found unsaturated firn on several ice shelves until 2200, but for the A1B scenario which has a lower radiative forcing than SSP5-8.5 (Collins et al., 2013). Under SSP2-4.5, we find that hydrofracturing conditions are likely not met before 2100 for 21 of the 56 monitored ice shelves (Fig. 12).





**Likely emergence of runoff conditions necessary for hydrofracturing over 1850-2200**



**Figure 10.** Time intervals at which the likely (17–83[th] percentiles) surface runoff production likely exceeds the threshold that makes hydrofracturing possible (see text), for 56 ice shelves or groups of ice shelves. The first time range, in italic, corresponds to SSP1-2.6, while the second range corresponds to SSP5-8.5. The years until 2100 are based on the weighted 16-model ensemble. The runoff likely range after 2100 was defined from eight unweighted models, from halfway between the 1[st] and 2[nd] models to halfway between the 5[th] and 6[th] models in order to account for both the diversity of model behaviours and ECS. The background map shows the eight-model mean runoff anomaly at the end of the 22[nd] century, with topography contours in grey (every 1000 m) and ice shelves delineated in thin black. The contours of the drainage basins of individual ice shelves is in thick black (from Mouginot et al., 2017 and Rignot et al., 2019).







**Figure 11.** Same as Fig. 10 but for the very likely range (5–95[th] percentiles) of the weighted 16-model runoff projections. Here the years after 2100 are estimated as the minimum and maximum of the eight models available until 2200.





**Figure 12.** Similar to Fig. 10 but for the SSP2-4.5 scenario. The dates in italic give the likely range and the others give the very likely range.



## 4   Discussion

In this study, we have used a relatively simple model to emulate RCM simulations. It is based on exponential fits for snow melt and accumulation rates. Although the method gives reasonably good results, the quality of melt rates exponential fits declines in a much warmer climate, typically after 2150 under SSP5-8.5. We suggest that more complex fitting methods, in particular through deep learning algorithms fed by multiple CMIP model variables (e.g., Sellevold and Vizcaino, 2021; van der Meer et al., 2023), may improve our approach. A complex algorithm trained on regional climate simulations with complex physical

parameterisations may be able to represent aspects that are not accounted for in our simple statistical–physical model, such as the emergence of bare ice and its effect on albedo, rainfall, the elevation feedback, or radiative feedbacks associated with the evolution of the cloud phase.

In this work, we have also assumed an instantaneous saturation of the firn beyond certain melt rates, while it can take more than a decade to saturate it if melt rates are just above the threshold (e.g., Donat-Magnin et al., 2021). This could be addressed

by introducing the temporal evolution of the depleted firn air volume in our simple model or in the deep learning approach, or in a simpler way, by introducing some time lag in the melt-runoff relationship.

Our approach has consisted of emulating MAR simulations. Other RCMs, possibly combined to elaborated firn models, have similar skills in representing typical Antarctic conditions (Mottram et al., 2021), but there is likely a considerable spread in their response to surface warming. For example, the depth and vertical resolution of firn models probably make important

differences in the timing of runoff production. One of the next priorities will therefore be to emulate the diversity of RCM sensitivities, which would make the uncertainty ranges much more comprehensive.

Here we have weighted the CMIP models to represent the very likely ECS distribution. We consider this as an improvement compared to unweighted multi-model means, but alternative weighting approaches should be explored, such as weights based on misfits with observational trends. Given the importance of internal variability for conditions at the surface of ice shelves

(Tsai et al., 2020), it would nonetheless be important to use multiple members of a given CMIP model. We therefore suggest that our method should be used to emulate all the ensemble members of individual CMIP experiments, which has not been explored in this study.

Last but not least, our estimation of the dates when runoff production becomes prone to hydrofracturing was based on a runoff threshold of 100 $\mathrm{kg\,m^{-2}\,yr^{-1}}$ over the ice shelf. This was motivated by the estimates of 200-300 $\mathrm{kg\,m^{-2}\,yr^{-1}}$ over

Larsen B prior to its collapse, suggesting the need for a smaller or equal threshold. All the results presented in this paper are based on a threshold of 100 $\mathrm{kg\,m^{-2}\,yr^{-1}}$, which is an empirical choice. Decreasing the threshold to 50 $\mathrm{kg\,m^{-2}\,yr^{-1}}$ shifts the dates by less than 10 years in the past for half of the ice shelves, but makes pre-industrial conditions favorable for 15% more ice shelves. Increasing the threshold to 150 $\mathrm{kg\,m^{-2}\,yr^{-1}}$ shifts the dates by less than 20 years in the future for half of the ice shelves. For some ice shelves, there can nonetheless be several decades of differences, again indicating that these dates are

more indications than real projections.



## 5 Conclusions

We have presented a novel mixed statistical-physical approach to emulate the spatio-temporal variability of the Antarctic Ice Sheet surface mass balance and runoff of the MAR regional climate model. We have presented evidence that this method can be used to extend existing MAR simulations to other periods and scenarios that were not originally processed through MAR. Our method is also able to construct pseudo-MAR simulations driven by CMIP models that have never been used to actually drive MAR simulations.

Our method is useful to populate ensembles of surface mass balance and runoff which are needed to constrain ice sheet model ensembles and to estimate the likely range of future sea level rise. We recommend using this type of approach to quickly derive surface conditions based on the next generation of CMIP models. This could be useful for the early stage of the upcoming ISMIP exercise, both for early initial ice sheet simulations and for a better selection of the CMIP models used to drive these simulations. We nonetheless encourage running more RCM experiments, in particular using other RCMs than MAR and RCMs driven by scenarios going beyond 2100.

We have used such a populated ensemble to correct the likely ECS distribution which was poorly represented by the CMIP6 ensemble. We find a likely SMB contribution to sea level rise of 0.4 to 2.2 cm from 1900 to 2010, and -3.4 to -0.1 cm from 2100 to 2099 in SSP1-2.6, versus -4.4 to -1.4 cm in SSP2-4.5 and -7.8 to -4.0 cm in SSP5-8.5. Based on a more limited and unweighted ensemble, we find a considerable uncertainty in the SMB contribution to sea level from 2000 to 2200: between -10 and -1 cm for SSP1-2.6 and between -33 and +6 cm for SSP5-8.5.

Given that our methodology also provides the spatio-temporal variability of runoff, we have then defined a criteria to identify the emergence of surface conditions prone to hydrofracturing. The emergence of such conditions over the historical period and in the near future qualitatively matches with observations of melt ponds and aquifers on a number of ice shelves. While a majority of ice shelves could remain safe from hydrofracturing in the SSP1-2.6 scenario, we estimate that all the Antarctic ice shelves will very likely be prone to hydrofracturing before 2130 in the SSP5-8.5 scenario. In combination to the ice shelf mechanical weakening induced by ocean warming, increased surface runoff is a major threat for the Antarctic ice shelves and for the grounding ice sheet outflows that they currently restrain.

*Code and data availability.* The tools used to extend the RCM data are available on https://github.com/nicojourdain/extend_SMB_melt_runoff.

*Author contributions.* NCJ designed the overall study, developed the statistical-physical method, made the analyses and wrote the initial draft. CA and CK produced the MAR simulations. GD, CA and CK discussed the results and contributed to the manuscript.

*Competing interests.* Nicolas Jourdain is an editor of The Cryosphere.



*Acknowledgements.*   This work was funded by the European Union's Horizon 2020 research and innovation programme under grant agreements No 869304 (PROTECT) and No 101003826 (CRiceS), and by the French National Research Agency under grant No ANR-22-CE01-0014 (AIAI). The work also benefited from the support of the French Government through the France 2030 program managed by ANR (ISClim, ANR-22-EXTR-0010). This publication is PROTECT contribution number XX.



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
