# Peer review of "Changes in Antarctic surface conditions and potential for ice shelf hydrofracturing from 1850 to 2200"

_EGUsphere, 2024_

## Referee Comment (RC1)

Changes in Antarctic surface conditions and potential for ice shelf hydrofracturing from 1850 to 2200

Jourdain et al. 2024

In this paper, the authors present a reconstruction method to emulate Antarctic surface mass balance conditions from Regional Climate Models from 1850 to 2200. They use this method to assess which ice shelves may be vulnerable to hydrofracture under different future emission scenarios. Developing methods to emulate important ice shelf processes that are under-resolved in Earth System Models is important for ice sheet modelling. However, I find the methods presented in this work to be extremely hard to follow, making it difficult to assess the robustness of this approach and the corresponding results. I believe that substantial clarification and elaboration throughout the methodology are necessary.

**Major concerns and general comments**

1)      The distinction between "meltwater production" and "runoff" is unclear throughout the paper. This is especially the case as the authors investigate both runoff as a contribution to sea level changes and the "emergence of runoff conditions necessary for hydrofracturing". I find these two outcomes of 'runoff' a bit contradictory because if meltwater runs off into the ocean and contributes to sea level rise, then it can't induce hydrofracture events, which result from the pressure of ponded meltwater (i.e. Bell et al 2018). I think the authors should be careful to clarify this terminology throughout the paper. For example, in L27 the authors state: "The exact warming level needed to trigger important production of runoff on a given ice shelf depends on the amount of snowfall and on the snow/firn temperature and density (Donat-Magnin et al 2021, van Wessem et al., 2023)" However, these studies specifically look at *meltwater production*, which is the important component for hydrofracture on ice shelves, not runoff.

2)   Methodology
There are many parts of the methodology that are not well explained and remain unclear. This makes it challenging to assess the robustness of the methodology and therefore the results. I suggest a graphical figure detailing the methodology. Below are some specific sentences or sections I did not understand or believe require additional detail:
   a)   L109: "To extend surface variables to a given local warming or cooling level, we always start from 20 different years (i.e. different values of *Tref*), then we average the 20 extended values." What does this mean? Does this essentially create a smoothed reconstruction?
   b)   L114: "The *a* and *b* parameters are obtained through a least-mean-square-fitting of an exponential curve for SMB minus runoff on the one hand and the surface melt rate of the other hand." A supplementary figure off this exponential curve would be very helpful.
   c)   L115: "The fit is done on the original model grid as regridding does not preserve exponential relationships."  I find this statement to be concerning. My understanding is that the exponential relationship between the two variables may weaken in the regridding but should remain? Further, if the exponential relationship parameters are fit on the original grid, and this relationship is not preserved in the regridding, is it then appropriate to apply this fit on the regridded data?
   d)   I am unsure how to interpret the *r* parameter (Eq. 3, 4, L121-124). Is this the percent of excess meltwater production that is converted to runoff (as opposed to that which ponds or refreezes)? If so, I expect that this value might be different on the grounded ice sheet vs ice shelves due to due to higher slopes on the grounded ice sheet.

e) Section 2.2.4: It is unclear how you reconstruct a cooler scenario from a warmer one. Do you use SSP5 to reconstruct SSP1? Or use warmer years as a reference time and reconstruct back in time? Also, what is the purpose of this and how will this be useful?

f) L157: "Similarly as in the previous subsection, each reconstructed year is the average of 20 reconstructions from a reference ranging from 10 years before to 9 years after the reconstructed year." I don't fully understand this sentence and have had to read it several times. It is perhaps related to the point mentioned in a) above?

g) Section 2.2.5: A graphic or schematic outlining the workflow here would be extremely helpful because I don't understand how the emulation is done. Additionally, Figure 3 is not very intuitive for me and should be better explained as I am unsure how to interpret it.

h) Figure 4: In panels e and f, the reconstructed runoff anomaly is too low, despite the melt anomaly (panel c and d) being fairly accurate for ice shelves and too high over the grounded ice sheet. Does this suggest some mis-parameterization in your method? Perhaps the wrong $r$ value?

i) L205-208: For these seven simulations... apply a ramping transition between the two methods from 2101 to 2120." I don't follow what is being done here and again, I think a figure or something would help the reader understand the methodology.

j) L280: The choice of a 100kg/m2/yr runoff threshold for triggering hydrofracture seems extreme and is not well-defended in the text. This is 50-67% less than the average meltwater production estimated prior to the collapse of Larsen B (200-300 according to the text). How was this threshold "empirically" chosen? Was it just based on Larsen A/B?

k) L291: How do you define "likely" or "very likely"?

3) Figure 1

L148: It seems a bit of a stretch to say that the extension of the RCM simulation is suitable for 25 years over the grounded ice sheet.... Really, there is just one year anomalously high SMB year at ~2125. Otherwise, the original MAR simulation and the reconstruction have opposite trends, even during this first 25 years.

In general, Figure 1 is concerning for me. It seems that this reconstruction method cannot be applied in a warming climate. However, the authors do apply some sort of reconstruction to obtain the results in Figures 6-12. How were these reconstructions obtained when Figure 1 demonstrates issues for applying this method in a warming climate? How can we trust the results presented here in light of Figure 1?

4) Relation to previous studies

In general, this manuscript is lacking some references to and context within recent literature. For example, how do the results in section 3.3 add to or fit within the context of previous ice-shelf potential instability studies (i.e. van Wessem et al., 2023; Dunmire et al., 2024; Alley et al., 2018; Lai et al., 2020)).

Additionally, some reverences to previous AIS SMB studies are missing (e.g., Gorte et al., 2020, Noel et al 2023).

5) Finally, the motivation for this work, and specifically how this method could be used in the context of ISMIP7, should be elaborated.

**Minor comments**

L7: "After correcting the distribution of equilibrium climate sensitivity of 16 climate models..." From just the abstract, it is unclear what this means.

L7: "… we find a likely contribution of surface mass balance to sea level rise of 0.4 to 2.2 cm from 1900 to 2010…". It does not make sense that the contribution of SMB to SLR would be positive for this period so I'm assuming this is with respect to a reference period? Same for the SLR contribution ranges in the following lines?

L25: "Hydrofracturing may strongly enhance the contribution of upstream glaciers to sea level rise." This is a bit misleading. The papers cited (among other work) indicate that *the removal/collapse of ice-shelves (perhaps due to hydrofracture events) causes a speed-up of upstream glaciers*, not just the hydrofracture event itself.

L41: "Because of these difficulties, only… which is generally insufficient to sample the CMIP model diversity." The "- when produced -" in this sentence threw me off a bit and I had to read it a few times to understand what was being said.

L43: "… correct unrealistic Equilibrium Climate Sensitivity…" A brief explanation for this concept would be helpful here.

L45: "Over the years, Antarctic Ice Sheet modellers have often scaled their best estimates of present-day accumulation to temperature anomalies from the CMIP models…)". This sentence fragment is unclear to me.

L67: "The surface mass balance and melting… Donat-Magnin et al (2020) and Kittel et al (2021)." I think a brief explanation of the results of these papers would be helpful here. I am left wondering: And how does MAR do in comparison to observational products?

L101: Assuming that all precipitation is entirely made of snow is a big assumption to make, especially for projections that extend to 2200 in high-emission scenarios. The impact of this assumption should at least be discussed somewhere in the paper.

L132-134: Should this really be interpreted as a 'mass loss rate' if Figure 1 shows anomaly values with respect to a reference period? The SMB for the reference period is positive (although the specific reference period value from MAR should be mentioned somewhere in the paper). Even though the line in Figure 1b decreases throughout the timeseries, mass loss doesn't occur until it reaches the negative magnitude of the reference period. For example, if the reference period SMB for ice-shelves ~500 Gt/yr, then surface mass loss doesn't really occur until approximately 2120 (when the time series reaches -500 Gt/yr).

L138: "… although this is still an improvement compared to the original IPSL-CM6A-LR outputs". I think it would be very interesting to have these original ESM timeseries plotted in Figure 1 as well.

L123: "… covering the aforementioned range, i.e. 0.5 to 0.9." This range is different from that mentioned before (0.6-0.85, L98).

Section 2.2.3: Somewhere in this section it should be clarified that Equation 4 is used to do this reconstruction.

L179-181: "The realistic SMB reconstructions derived from MAR-ACCESS1.3 are mostly compensations between overestimated melt and overestimated accumulation". Why do you say this?

L210: "… with a 20-year transition." What does this mean?

L238: Figure 8 is mentioned before Figure 7

L253: What does "weaker SMB" mean?

L265: "Spatially, a net surface mass loss arises for several ice shelves…" I find this to be a bit misleading since Figure 7 shows SMB anomalies from a reference period, not absolute SMB. Is there actually a net surface mass loss or is it just lower than the reference period?

L294 – It should be mentioned that George VI ice shelf has compressive stresses which do not promote hydrofracture occurance (Labarbera et al., 2011)

L 311: What is the A1B scenario?

**Technical corrections**

L33: progresses → progress

L62: Citation needed for the pore close-off density.

L180: "ooverestimated"

L185: "emulation" → emulations

L234: "scenario" → "scenarios"

L 254: End parenthesis after "(Fig. 9."

L262: "to the exception" → "with the exception"

Figure 8 caption: "same as Fig. ?? is also shown"

**References used in this review**

Bell, R.E., Banwell, A.F., Trusel, L.D. *et al.* Antarctic surface hydrology and impacts on ice-sheet mass balance. *Nature Clim Change* **8**, 1044–1052 (2018). https://doi.org/10.1038/s41558-018-0326-3

Lai, CY., Kingslake, J., Wearing, M.G. *et al.* Vulnerability of Antarctica's ice shelves to meltwater-driven fracture. *Nature* **584**, 574–578 (2020). https://doi.org/10.1038/s41586-020-2627-8

van Wessem, J.M., van den Broeke, M.R., Wouters, B. *et al.* Variable temperature thresholds of melt pond formation on Antarctic ice shelves. *Nat. Clim. Chang.* **13**, 161–166 (2023). https://doi.org/10.1038/s41558-022-01577-1

Dunmire, D., Wever, N., Banwell, A.F. *et al.* Antarctic-wide ice-shelf firn emulation reveals robust future firn air depletion signal for the Antarctic Peninsula. *Commun Earth Environ***5**, 100 (2024). https://doi.org/10.1038/s43247-024-01255-4

Alley, K. E., Scambos, T. A., Miller, J. Z., Long, D. G. & MacFerrin, M. Quantifying vulnerability of Antarctic ice shelves to hydrofracture using microwave scattering properties. *Remote Sens. Environ.* **210**, 297–306 (2018).

Gorte, T., Lenaerts, J. T. M., and Medley, B.: Scoring Antarctic surface mass balance in climate models to refine future projections, The Cryosphere, 14, 4719–4733, https://doi.org/10.5194/tc-14-4719-2020, 2020.

Noël, B., van Wessem, J.M., Wouters, B. *et al.* Higher Antarctic ice sheet accumulation and surface melt rates revealed at 2 km resolution. *Nat Commun* **14**, 7949 (2023). https://doi.org/10.1038/s41467-023-43584-6

Labarbera, C. H. & Macayeal, D. R. Traveling supraglacial lakes on George VI Ice Shelf. *Antarctica* **38**, 1–5 (2011).

---

## Author Comment (AC1)

Response to anonymous Referee #1.

We thank the referee for the time spent on our manuscript and for these comments. We have realised that several aspects of our methodology were not so clearly explained and we believe that these comments will greatly improve the clarity of our manuscript. We believe that most of the points raised by the referee can be addressed by clarifying our method and we hereafter provide more details on how we will address these comments.

1) The distinction between "meltwater production" and "runoff" is unclear throughout the paper. This is especially the case as the authors investigate both runoff as a contribution to sea level changes and the "emergence of runoff conditions necessary for hydrofracturing". I find these two outcomes of 'runoff' a bit contradictory because if meltwater runs off into the ocean and contributes to sea level rise, then it can't induce hydrofracture events, which result from the pressure of ponded meltwater (i.e. Bell et al 2018). I think the authors should be careful to clarify this terminology throughout the paper. For example, in L27 the authors state: "The exact warming level needed to trigger important production of runoff on a given ice shelf depends on the amount of snowfall and on the snow/firn temperature and density (Donat-Magnin et al 2021, van Wessem et al., 2023)" However, these studies specifically look at meltwater production, which is the important component for hydrofracture on ice shelves, not runoff.

⇒ We agree that it was misleading and we will better define the quantities in the revised manuscript.

Runoff into the ocean is a negative contribution to the surface mass balance. It is produced if surface melt and/or rain rates are high enough (i) to bring the temperature of underlying snow and firn layers to the freezing point and (ii) to percolate and saturate the pore space in the snow and firn layers, which is sometimes referred to as firn air depletion (Pfeffer et al., 1991; Kuipers Munneke et al., 2014; Alley et al., 2018; Donat-Magnin et al., 2021). Surface melt and/or rainfall beyond the pore space saturation do not necessarily lead to runoff into the ocean. Liquid water in excess can alternatively be transported horizontally and stay on the ice shelf and form ponds. In some circumstances, these processes can trigger ice-shelf collapse through hydrofracturing (Bell et al., 2018; Robel and Banwell, 2019; Lai et al., 2020).

The MAR model does not represent the horizontal transport of meltwater within the firn or at the surface of ice shelves, but assumes that the meltwater in excess for individual snow/firn column is removed from the system, which is why it is usually referred to as "runoff" in this model (e.g., Agosta et al., 2019; Kittel et al. 2021). We will reword "runoff" as "liquid water in excess" when this is relevant throughout the manuscript.

2) Methodology
There are many parts of the methodology that are not well explained and remain unclear. This makes it challenging to assess the robustness of the methodology and therefore the results. I suggest a graphical figure detailing the methodology.

⇒ This is a good suggestion; we will make a graphical summary of our methodology in the revised version.

Below are some specific sentences or sections I did not understand or believe require additional detail:

a) L109: "To extend surface variables to a given local warming or cooling level, we always start from

20 different years (i.e. different values of Tref), then we average the 20 extended values." What does this mean? Does this essentially create a smoothed reconstruction?

⇒ We will better explain this part in the revised manuscript. To reconstruct the annual melt rate (or SMB minus runoff) at a given location and on a given year, an option is to start from the annual melt rate in another simulation and to correct this value to account for the air temperature difference (ΔT) between the two simulations. Given internal climate variability and uncertainty in the exponential fits, there is no reason to only use 1986 from simulation A to reconstruct 1986 in simulation B. So we reconstruct 1986 in simulation B twenty times, for years between 1976 and 1995 (and different ΔT values for every reconstruction) in simulation A, then we take the average of these 20 reconstructions.

We process similarly when we extend a simulation back in time: for example, 1950 (or any other year before 1980) is reconstructed 20 times from individual years in the corresponding MAR simulation that covers 1980–1999, and then we take the average.

This removes a part of the internal climate variability in the melt or SMB pattern of a given year (e.g., the influence of a strong synoptic event at a given year) but keeps the internal climate variability of air temperature in the reconstructed timeseries.

b) L114: "The a and b parameters are obtained through a least-mean-square-fitting of an exponential curve for SMB minus runoff on the one hand and the surface melt rate of the other hand." A supplementary figure off this exponential curve would be very helpful.

⇒ This will be added.

c) L115: "The fit is done on the original model grid as regridding does not preserve exponential relationships." I find this statement to be concerning. My understanding is that the exponential relationship between the two variables may weaken in the regridding but should remain? Further, if the exponential relationship parameters are fit on the original grid, and this relationship is not preserved in the regridding, is it then appropriate to apply this fit on the regridded data?

⇒ Yes, in principle, regridding weakens rather than removes the exponential relationship, but we will remove this statement because the simulations were interpolated conservatively from a 35 km grid to a 4 km grid, so the values on the interpolated grid are almost the same as on the original grid.

d) I am unsure how to interpret the r parameter (Eq. 3, 4, L121-124). Is this the percent of excess meltwater production that is converted to runoff (as opposed to that which ponds or refreezes)? If so, I expect that this value might be different on the grounded ice sheet vs ice shelves due to due to higher slopes on the grounded ice sheet.

⇒ This will be clarified with a better definition of runoff and liquid water in excess in a given firn column. The r parameter accounts for the fact the meltwater first needs to deplete the firn air content before being available for potential ponding (this is well defined in Donat-Magnin et al. 2021 and van Wessem et al., 2023). If there is a lot of snowfall, all meltwater ends up refreezing in the firn. If snowfall is low, it is easier to saturate the firn with meltwater (i.e., deplete the firn air content), after which any additional meltwater is potentially available for ponding. Therefore, if the melt to snowfall ratio exceeds r, it just means that there is locally a potential for ponding and hydro-fracturing.

We do not simulate the horizontal transport of meltwater. Nevertheless, when we explore the hydrofracturing potential at the end of the paper, we average quantities at the scale of an ice shelf so that our estimates are less affected by how meltwater has moved at the surface. Furthermore, we account for the sloping grounded basin upstream of the ice shelf by assuming that the meltwater in excess over the grounded ice flows towards the ice shelf.

e) Section 2.2.4: It is unclear how you reconstruct a cooler scenario from a warmer one. Do you use SSP5 to reconstruct SSP1? Or use warmer years as a reference time and reconstruct back in time? Also, what is the purpose of this and how will this be useful?

⇒ As written in section 2.2.4, "we evaluate the reconstructions of both SSP1-2.6 and SSP2-4.5 from SSP5-8.5". We will indicate the purpose of this in section 2.2.4, which is currently only shown in sections 3 and 4. This basically allows to have a large ensemble of SMB/melt projections based on a limited number of regional climate model (RCM) simulations. RCM simulations are numerically expensive and require 6-hourly CMIP model outputs which are not always available. So based on a single RCM simulation under SSP5-8.5, it is possible to emulate any colder scenario over the same period.

f) L157: "Similarly as in the previous subsection, each reconstructed year is the average of 20 reconstructions from a reference ranging from 10 years before to 9 years after the reconstructed year." I don't fully understand this sentence and have had to read it several times. It is perhaps related to the point mentioned in a) above?

⇒ Yes, see previous response. This will be clarified through the graphical summary.

g) Section 2.2.5: A graphic or schematic outlining the workflow here would be extremely helpful because I don't understand how the emulation is done. Additionally, Figure 3 is not very intuitive for me and should be better explained as I am unsure how to interpret it.

⇒ Yes, we will summarise the workflow in a graphical summary.

We do not really understand what to improve in the description of Fig. 3. The caption is quite explanatory, and this way of presenting results is also found in other papers (e.g., Figs. 4 & 7 in Barthel et al., 2020). Each diagonal direction represents one CMIP model, in black for the original MAR simulation forced by this CMIP model, and in colour for the emulation from another MAR simulation forced by a different CMIP model. The emulations from a given CMIP model are represented by small circles of the same colour and are linked together with a line to give a better overview of the general behaviour.

h) Figure 4: In panels e and f, the reconstructed runoff anomaly is too low, despite the melt anomaly (panel c and d) being fairly accurate for ice shelves and too high over the grounded ice sheet. Does this suggest some mis-parameterization in your method? Perhaps the wrong r value?

⇒ No, the reconstructed melting runoff anomalies are both slightly too high in Fig. 4, as indicated by the positive bias values indicated in brackets for panels c, d, e, f. If the referee was thinking about Fig. 2 and not Fig. 4, there is indeed a positive bias in the melt anomaly and a negative bias in the runoff (i.e., excess of liquid water beyond firn

saturation). Decreasing the r value would clearly reduce the runoff bias in this figure, but not in others. This indeed shows that the parameterisation is not perfect. We will add a comment on this.

i) L205-208: For these seven simulations... apply a ramping transition between the two methods from 2101 to 2120." I don't follow what is being done here and again, I think a figure or something would help the reader understand the methodology.

⇒ Yes, this will be clarified in the graphical summary. There are two ways to emulate a simulation beyond 2100: extending from 2081-2100 (method described in section 2.2.3) and reconstructing from another simulation that covers 2101-2200 (here IPSL-CM6-LR--SSP5-8.5; method described in section 2.2.4 for other scenarios of IPSL-CM6-LR or 2.2.5 for other models). The extension from 2081-2100 allows some continuity with the existing 1980-2100 simulation but its fidelity decreases with time. Therefore, we do a linear combination of the 2 reconstructions from 2101 to 2120, with a weight of the extension from 2081-2100 that decreases linearly from 1 to 0 over the 20 years.

j) L280: The choice of a 100kg/m2/yr runoff threshold for triggering hydrofracture seems extreme and is not well-defended in the text. This is 50-67% less than the average meltwater production estimated prior to the collapse of Larsen B (200-300 according to the text). How was this threshold "empirically" chosen? Was it just based on Larsen A/B?

⇒ We can just tell that Larsen B was over the threshold when it collapsed, not just at the threshold given that mechanical conditions also need to be satisfied. Therefore, we took a threshold value smaller than the estimated production of meltwater beyond firn saturation at the time of collapse. Here we emphasize that this is not the meltwater production, this is the excess of meltwater beyond firn saturation, so any $kg\ m^{-2}\ yr^{-1}$ is potentially available for hydrofracturing. In comparison, van Wessem et al. (2023) consider that melt ponds form when the melt over accumulation ratio exceeds r=0.7, i.e. a firn saturated with meltwater, which is equivalent to having a threshold just above zero (we will mention this in the revised manuscript). There is of course some sensitivity to this threshold, as written in our Discussion section:

*"Last but not least, our estimation of the dates when runoff production becomes prone to hydrofracturing was based on a runoff threshold of 100 $kgm^{-2}\ yr^{-1}$ over the ice shelf. This was motivated by the estimates of 200-300 $kg\ m^{-2}\ yr^{-1}$ over Larsen B prior to its collapse, suggesting the need for a smaller or equal threshold. All the results presented in this paper are based on a threshold of 100 $kg\ m^{-2}\ yr^{-1}$, which is an empirical choice. Decreasing the threshold to 50 $kg\ m^{-2}\ yr^{-1}$ shifts the dates by less than 10 years in the past for half of the ice shelves, but makes pre-industrial conditions favorable for 15% more ice shelves. Increasing the threshold to 150 $kg\ m^{-2}\ yr^{-1}$ shifts the dates by less than 20 years in the future for half of the ice shelves. For some ice shelves, there can nonetheless be several decades of differences, again indicating that these dates are more indications than real projections."*

k) L291: How do you define "likely" or "very likely"?

⇒ We use these terms as in the IPCC reports where they indicate the assessed likelihood of an outcome or a result: virtually certain 99-100% probability, very likely 90-100% probability, likely 66-100% probability. We agree that this needs to be better defined. Currently it is only defined in the caption of Tab. 4 in terms of percentiles "(likely

range, i.e., 17–83th percentile) [very likely range, i.e., 5–95th percentile]". This will be clarified earlier in the revised manuscript.

3) Figure 1
L148: It seems a bit of a stretch to say that the extension of the RCM simulation is suitable for 25 years over the grounded ice sheet…. Really, there is just one year anomalously high SMB year at ~2125. Otherwise, the original MAR simulation and the reconstruction have opposite trends, even during this first 25 years.
In general, Figure 1 is concerning for me. It seems that this reconstruction method cannot be applied in a warming climate. However, the authors do apply some sort of reconstruction to obtain the results in Figures 6-12. How were these reconstructions obtained when Figure 1 demonstrates issues for applying this method in a warming climate? How can we trust the results presented here in light of Figure 1?

⇒ Our intention is not to assess the detail of the interannual variability over these 25 years, but to describe the overall bias that could affect an ice-sheet model in the long term. We nonetheless understand the need for a more objective statement and we will mention that the reconstructed SMB has a relative bias smaller that 12% on average over 2101-2125 for the grounded ice sheet and over 2101-2150 for the ice shelves (for r=0.5 and r=0.6).

We will also better emphasize in the text that this method is not suitable for extensions to a much warmer climate. In the final reconstructions used in section 3, we only use this method over a 20-year ramp down transition (2101–2120) as explained previously, and to extend the 1980–1999 period to colder conditions (1850–1979) which is less problematic. The main reconstruction of 2101–2200 for all models and scenarios is based on the MAR–IPSL-CM6A-LR simulation that covers 2101–2200.

4) Relation to previous studies
In general, this manuscript is lacking some references to and context within recent literature. For example, how do the results in section 3.3 add to or fit within the context of previous ice-shelf potential instability studies (i.e. van Wessem et al., 2023; Dunmire et al., 2024; Alley et al., 2018; Lai et al., 2020)).

Additionally, some reverences to previous AIS SMB studies are missing (e.g., Gorte et al., 2020, Noel et al 2023).

⇒ That's a fair point, we will add more comparisons to similar types of predictions.

Van Wessem et al. (2023) assume r=0.7 which was calculated by Pfeffer (1991) for a particular snow density and temperature. As Donat-Magnin et al. (2021), we find that slightly smaller values better match with the snow firn properties simulated by MAR in Antarctica. Van Wessem et al. (2023) then used air temperatures from the CMIP6 ensemble to extrapolate melt and accumulation based on a similar fitting method as the one used in our article (which is also similar to Donat-Magnin et al., 2021).

Dunmire et al. (2024) was published one and a half months after our submission. It has some similarities with our study, so we will definitely include a comparison to their results in the revised manuscript. They emulate firn air content which is exactly the same as the liquid water in excess that we emulate to estimate conditions prone to hydrofracturing, and they also use air temperature from the CMIP6 models, but the detail of their emulation and the training database are very different from our study.

We believe that most original aspects of our approach compared to these studies are (i) the time coverage back to 1850 and until 2200 while other studies cover ~1980-2100, (ii) the weighted ensemble members to have a more realistic representation of the equilibrium climate sensitivity than the raw CMIP6 data, (iii) the relatively large number of MAR simulations used to assess and calibrate our simple emulator.

In terms of results, our findings are in line with previous results: the Getz, Ross and Ronne-Filchner ice shelves do not reach melt conditions prone to hydrofracturing before 2100 whatever the emission scenario, while the ice shelves in the Peninsula and East Antarctica easily reach melting conditions prone to hydrofracturing before 2100 (Alley et al., 2018; Donat-Magnin 2021; van Wessem et al. 2023; Dunmire et al., 2024). As van Wessem et al. (2023), we estimate that Shackleton, Amery, Roi Baudoin and Larsen C ice shelves are prone to hydrofracturing in all scenarios before 2100.

Regarding the other articles mentioned by the reviewer:

- Lai et al. (2020) is already mentioned and provides the description of the mechanical conditions that can lead to hydrofracturing in the presence of meltwater beyond firn saturation.

- Noel et al. (2023) was published just one month before our submission so we did not have time to include it but we will do it in the revised manuscript. The aim of their statistical approach is to provide SMB and melt rates at a higher horizontal resolution than the original regional climate model. We will suggest that future work combines their approach and our approach to emulate large ensemble of SMB and melt rate projections.

- Gorte et al. (2020) provide an interesting evaluation of SMB in the CMIP5 and CMIP6 models, but assume that SMB can be approximated as precipitation minus evaporation/sublimation, i.e., they assume that runoff is negligible, which is not a good approximation beyond 2100 in the warmest scenarios.

5) Finally, the motivation for this work, and specifically how this method could be used in the context of ISMIP7, should be elaborated.

⇒ We will better explain how all this work can be useful to ISMIP7 or similar ice sheet projections. This reconstruction method is already used to provide the surface mass balance (SMB) in the ice-sheet multi-model projections of the PROTECT European project (https://doi.org/10.5194/egusphere-egu24-17095). In addition, our reconstructed melting beyond firn saturation could also be used to feed calving parameterisations or to impose dates of collapse as was done in ISMIP6.

The idea is that the CMIP models usually don't have a good snow/melt physics (some like IPSL don't represent the fate of meltwater in the firn), so that it is preferable to use regional climate models that were developed for polar regions (e.g., MAR, RACMO). However, the weakness of these regional models is the associated requirement for additional skills, additional inputs (e.g., 6-hourly 3-dimensional fields) and processing/computing time. Thanks to the emulation, we were able to provide the SMB

for multiple CMIP models and scenarios beyond 2100 (which are difficult to process through a regional climate model given that the 6-hourly CMIP outputs were not saved after 2100 except for IPSL-CM6A-LR--SSP5-8.5 on our request). We were also able to reconstruct the SMB for CMIP models that were never processed through any regional climate models, which increases the diversity of SMB projections. Furthermore, our method can also be used to reconstruct SMB back to 1850 a period for which we currently have no regional climate simulations.

Minor comments

L7: "After correcting the distribution of equilibrium climate sensitivity of 16 climate models…" From just the abstract, it is unclear what this means.

⇒ We believe that the concept of "equilibrium climate sensitivity" is sufficiently known in the climate community to appear in the abstract, but we will define it in the text.

L7: "… we find a likely contribution of surface mass balance to sea level rise of 0.4 to 2.2 cm from 1900 to 2010…". It does not make sense that the contribution of SMB to SLR would be positive for this period so I'm assuming this is with respect to a reference period? Same for the SLR contribution ranges in the following lines?

⇒ Thank you, the entire sentence was wrong and will be replaced with:
"we find a likely contribution of surface mass balance to sea level rise of **-2.2 to -0.4** cm from 1900 to 2010, and -3.4 to -0.1 cm from **2000** to 2099 under the SSP1-2.6 scenario, versus -4.4 to -1.4 cm under SSP2-4.5 and -7.8 to -4.0 cm under SSP5-8.5".

L25: "Hydrofracturing may strongly enhance the contribution of upstream glaciers to sea level rise." This is a bit misleading. The papers cited (among other work) indicate that the removal/collapse of ice-shelves (perhaps due to hydrofracture events) causes a speed-up of upstream glaciers, not just the hydrofracture event itself.

⇒ We will replace "hydrofracturing" with "ice shelf collapse resulting from hydrofracturing" in this sentence.

L41: "Because of these difficulties, only… which is generally insufficient to sample the CMIP model diversity." The "- when produced –" in this sentence threw me off a bit and I had to read it a few times to understand what was being said.

⇒ Ok, we will remove this part of the sentence.

L43: "… correct unrealistic Equilibrium Climate Sensitivity…" A brief explanation for this concept would be helpful here.

⇒ Yes, we will add "The ECS is the long-term increase in global mean surface temperature after a doubling of $CO_2$ concentrations. It can be used to characterise the sensitivity of individual climate models to changes in the radiative forcing."

L45: "Over the years, Antarctic Ice Sheet modellers have often scaled their best estimates of present-day accumulation to temperature anomalies from the CMIP models…". This sentence fragment is unclear to me.

⇒ We will replace with "Over the years, Antarctic Ice Sheet modellers have often scaled their best estimates of present-day accumulation to temperature anomalies from the CMIP models (e.g., based on the Clausius-Clapeyron relationship as in Gregory and Huybrechts, 2006)".

L67: "The surface mass balance and melting… Donat-Magnin et al (2020) and Kittel et al (2021)." I think a brief explanation of the results of these papers would be helpful here. I am left wondering: And how does MAR do in comparison to observational products?

⇒ In the revised manuscript, we will summarise the main results of the melt and SMB evaluations presented in these articles:
"Agosta et al. (2019) used firn-core SMB estimates to evaluate a MAR configuration covering the entire ice sheet: their SMB spatial pattern was well captured and the mean bias was 4%. Donat-Magnin et al. (2020) compared their MAR configuration of the Amundsen Sea sector to automatic weather stations, airborne-radar and firn-core SMB, melt days from satellite microwave, and melt rates from satellite scatterometer. They obtained good results for near-surface temperatures (mean overestimation of 0.1°C), near-surface wind speeds (mean underestimation of 0.42 ms$^{-1}$ and SMB (local biases lower than 20%). The mean surface melt rate over the Amundsen Sea region was underestimated by 18% but the interannual variability was well captured for both melt rate and the annual number of melting days. The aforementioned MAR simulations were forced by atmospheric reanalyses and Kittel et al. (2021) showed that the Antarctic SMB simulated by MAR forced by climate models was close to MAR forced by the ERA5 reanalysis over the recent decades".

L101: Assuming that all precipitation is entirely made of snow is a big assumption to make, especially for projections that extend to 2200 in high-emission scenarios. The impact of this assumption should at least be discussed somewhere in the paper.

⇒ This will be added.

Donat-Magnin et al. (2021) find that mean rainfall in 2081-2100 under the RCP8.5 scenario is still one to two orders of magnitude smaller than snowfall over ice shelves in the Amundsen Sea. The only regional climate simulation from MAR until 2200 (the one forced by IPSL-CM6A-LR--SSP5-8.5) does simulate the actual rainfall/snowfall distribution and still has a positive surface mass balance over ice shelves in 2200, which indicates that snowfall is still dominant over rainfall. Furthermore, as discussed in Appendix B of Donat-Magnin et al. (2021), a greater rainfall rate than melting rate is required to saturate the firn (because melting both removes snow and depletes its air content, while rainfall just depletes the air content). The emergence of rainfall nonetheless contributes to decrease the fidelity of our method towards the end of the 22$^{nd}$ century.

L132-134: Should this really be interpreted as a 'mass loss rate' if Figure 1 shows anomaly values with respect to a reference period? The SMB for the reference period is positive (although the specific reference period value from MAR should be mentioned somewhere in the paper). Even though the line in Figure 1b decreases throughout the timeseries, mass loss doesn't occur until it reaches the negative magnitude of the reference period. For example, if the reference period SMB for ice-shelves ~500 Gt/yr, then surface mass loss doesn't really occur until approximately 2120 (when the time series reaches -500 Gt/yr).

⇒ Yes, this will be reformulated. It is clearly shown in Fig. 8 that the current SMB over ice shelves is ~500 Gt/yr, so an anomaly of 2200 Gt/yr is just a negative SMB of 1700 Gt/yr (and an actual mass loss depends on basal melt, calving and the ice flux across the grounding line).

L138: "… although this is still an improvement compared to the original IPSL-CM6A-LR outputs". I think it would be very interesting to have these original ESM timeseries plotted in Figure 1 as well.

⇒ We agree, this will be added.

L123: "… covering the aforementioned range, i.e. 0.5 to 0.9." This range is different from that mentioned before (0.6-0.85, L98).

⇒ We will reformulate. The 0.5–0.9 range covers the 0.60–0.85 range discussed in previous studies.

Section 2.2.3: Somewhere in this section it should be clarified that Equation 4 is used to do this reconstruction.

⇒ This will be clarified as suggested.

L179-181: "The realistic SMB reconstructions derived from MAR-ACCESS1.3 are mostly compensations between overestimated melt and overestimated accumulation". Why do you say this?

⇒ Because the melt rates emulated from MAR-ACCESS1.3 are largely overestimated, as seen by the large green pentagon in Fig. 3c, which means that the realistic SMB is due to underestimated accumulation. This is to argue that emulations from MAR-ACCESS1.3 are outliers despite realistic emulated SMB over the grounded ice sheet.

L210: "… with a 20-year transition." What does this mean?

⇒ This will be clarified, see previous responses.

L238: Figure 8 is mentioned before Figure 7

⇒ This will be corrected in the revised manuscript.

L253: What does "weaker SMB" mean?

⇒ This will be replaced with "lower SMB".

L265: "Spatially, a net surface mass loss arises for several ice shelves…" I find this to be a bit misleading since Figure 7 shows SMB anomalies from a reference period, not absolute SMB. Is there actually a net surface mass loss or is it just lower than the reference period?

⇒ This will be reformulated. The anomaly is compared to the present-day SMB in Fig. 8.

L294 – It should be mentioned that George VI ice shelf has compressive stresses which do not promote hydrofracture occurrence (Labarbera et al., 2011)

⇒ We will mention this.

L 311: What is the A1B scenario?

⇒ This is a scenario that was used in CMIP3 and IPCC-AR4 which is cited in this sentence.

Technical corrections:

⇒ Thank you, these will be corrected.

L33: progresses à progress
L62: Citation needed for the pore close-off density.
L180: "ooverestimated"
L185: "emulation" à emulations
L234: "scenario" à "scenarios"
L 254: End parenthesis after "(Fig. 9."
L262: "to the exception" à "with the exception"
Figure 8 caption: "same as Fig. ?? is also shown"

---

## Author Comment (AC2)

Response to anonymous Referee #2.

We thank the referee for the time spent on our manuscript and for these careful and valuable comments.

General
This is an interesting paper that addresses an important problem: when can we expect runoff from Antarctic ice shelves, leading to mass loss and/or meltwater ponding which is commonly regarded as a precursor for ice shelf hydrofracture. The authors use multiple MAR simulations of future Antarctic climates, with which they calibrate a simple statistical model to emulate melt, runoff so as to approximate SMB and (instantaneous) firn saturation. Not considered are sublimation, rain and the time it takes to saturate the firn. When compared directly to MAR output, the emulator gives mixed results, yet provides valuable insights in the uncertainties that arise from the driving global models. The writing can be clarified in places (see below), the figures are generally clear.

**Major comments**

The abstract is at places confusing. It goes in one step from describing runoff emulation to AIS SLR contribution. Although it is stated that the mass loss numbers are based on SMB alone, the link that is made to hydrofracturing in the title and the presented numbers in terms of sea level rise could trick the reader into thinking that dynamical effects are also considered. It should be made clear from the outset is that this study treats runoff in two ways: as a source of mass loss and as a threshold to induce ponding and hydrofracturing (after which the dynamical effect on the ice sheet is not considered).

⇒ This is a good suggestion. We will improve the clarity of these aspects throughout the revised manuscript and in particular in the abstract.

Also in the abstract: "Based on a more limited and uncorrected ensemble, we find a considerable uncertainty in the contribution to sea level from 2000 to 2200". It is unclear whether this enhanced uncertainty derives from the fact that the ensemble is smaller and uncorrected, or from the method, or all of these. Please only include major results in the abstract, the details can be elaborated upon in the text.

⇒ In the description of our results, we will make it clearer that the larger uncertainty after 2100 is largely due to the diverging behavior of CMIP models: for some CMIP models, SMB over the grounded ice sheet remains over present-day values after 2100 while it strongly decreases in other models. For this reason, and given the limited sample size of 8 models, we do not provide an average after 2100 and only indicate the spread. The sentence quoted from the abstract will be rewritten as "*Based on a more limited and uncorrected ensemble, we find a considerable spread across CMIP models in the contribution to sea level from 2000 to 2200*".

l.59: The statistical model is based on MAR data, which is a fair choice given that this model has been used for multiple future runs. But for the statistical model to be useful, MAR must provide reliable runoff. Here it is stated that the atmosphere in MAR is coupled to a 30 layer snow model "...representing the first 20 m of snow/firn with refined resolution at the surface". But in many places the firn layer in Antarctica is considerably deeper than 20 m? What does this imply for runoff simulated by MAR? See also my comments on treatment of percolation and runoff evaluation below.

$\Rightarrow$ Only using MAR is certainly a caveat but this was the only RCM for which we had an ensemble of projections sufficiently large to be robustly emulated. This caveat was already acknowledged in the Discussion section. We nonetheless agree that simulating 100 m of firn would be better than 20 m, although the potential presence of ice lenses makes it difficult to anticipate the exact effect. The firn depth is expected to change the timing of firn saturation/air depletion, but not the threshold for firn saturation which is more related to the melting and snowfall rates. We have added the sentence in red to the relevant paragraph in the Discussion section:

"*Our approach has consisted of emulating MAR simulations. Other RCMs, possibly combined to elaborated firn models, have similar skills in representing typical Antarctic conditions (Mottram et al., 2021), but there is likely a considerable spread in their response to surface warming. For example, the depth and vertical resolution of firn models probably make important differences in the timing of runoff production. The 20-m firn layer simulated in MAR thus likely reaches liquid-water saturation earlier than models with a thicker firn layer. One of the next priorities will therefore be to emulate the diversity of RCM sensitivities, which would make the uncertainty ranges much more comprehensive.*"

l.65: " If liquid water is not able to percolate further down, then it fills the entire porosity space of surface layers, and the excess is considered as runoff and removed from the snow/firn model (there is no representation of ponds or horizontal routing)". From this description it is not clear to me how water percolation interacts with ice lenses in the firn layer and how the process of filling up works before runoff starts.

$\Rightarrow$ This sentence will be clarified so that the entire paragraph reads:

"*In the presence of surface melting or rainfall, liquid water percolates downward into the next firn layers with a water retention of 5% of the porosity in each successive layer. The firn layers are fully permeable until they reach a close-off density of 830 kg m$^{-3}$. To account for possible cracks in ice lenses and moulins, the part of available water that is transmitted downward to the next layer decreases as a linear function of firn density, from 100% transmitted at the close-off density to zero at 900 kg m$^{-3}$ and beyond. If liquid water is not able to percolate further down, it remains where it is. When the entire porosity space in the surface grid cell is filled with liquid water or if the surface grid cell is denser than 900 kg m$^{-3}$, any additional surface melt is considered as runoff and removed from the snow/firn model. There is no representation of ponds or horizontal routing.*"

l.67: "The surface mass balance and melting conditions produced by MAR have been evaluated in comparison to observational." Has there also been an attempt to evaluate the modelled meltwater ponding/runoff? I seem to remember that Van Wessem et al. (2023) compared their predicted ponding potential to satellite observations of melt ponds.

$\Rightarrow$ Van Wessem et al. (2023) indeed presented a comparison to an observational estimate of melt pond volume (derived from Sentinel 2). This comparison is qualitative as neither MAR nor RACMO simulate ponding (they remove the excess of liquid water and do not simulate horizontal transport of liquid water). We nonetheless show here the runoff produced by MAR forced by ERA5 over 2015-2022 in comparison the melt pond volume over the same period:

[Figure]

We find that the areas of high runoff in MAR generally correspond to the areas where high melt pond volumes are estimated from Sentinel 2 data, even if the area of high runoff over Larsen C is larger than the area of large melt pond volume in the satellite product. We will add a sentence on this overall qualitative agreement although we may not include the figure.

We also note that already included this statement at the end of section 3.3, which may be considered as a kind of evaluation:
"*In our projections, conditions favorable to hydrofracturing become likely by ~2050 for several East Antarctic ice shelves (Fig. 10). This is the case of Nivl, Roi Baudouin, Amery and Shackleton ice shelves on which widespread melt ponds or aquifers are already observed in present-day conditions (Kingslake et al., 2017; Bell et al., 2018; Stokes et al., 2019)*".

Figure 1 shows results for the integrated grounded ice/ice shelves only. How do the spatial patterns look?

⇒ This is a good suggestion. The revised manuscript will include the maps for some of the cases integrated in Figs. 1, 2, 4 (in Appendix).

**Minor comments**

l. 12: "Based on a runoff criteria.." Please note that 'criteria' is plural, so either use "Based on runoff criteria.." or "Based on a runoff criterion.."

⇒ Thank you, this will be corrected.

l. 12: "we identify the emergence of surface conditions prone to hydrofracturing" Suggest: "we identify the timing of surface conditions that make ice shelves prone to hydrofracturing" or something similar.

⇒ This will be corrected as suggested.

l. 13: " A majority of ice shelves could remain safe" Suggest to reformulate: "Our results suggest that the majority..."

⇒ This will be corrected as suggested.

l.16: precipitation -> snowfall

⇒ Precipitation consists of snowfall + rainfall, and rainfall is a positive term in the surface mass balance that effectively contributes to mass gain until the firn is saturated with liquid water or until the surface consists of bare ice. But rainfall has a minor effect and this first sentence may be clearer with "snowfall", so we will correct as suggested.

l. 18: However, if air temperatures exceed -> However, model simulations suggest that, if atmospheric warming exceeds

⇒ This will be corrected as suggested.

l. 22: can be conducive of -> can be conducive to

⇒ This will be corrected as suggested.

l. 31: sea level -> sea level change

⇒ This will be corrected as suggested.

l. 36: in particular with regard to firn saturation by meltwater and subsequent runoff -> in particular with regard to firn saturation by meltwater and subsequent ponding

⇒ This will be corrected as suggested.

l. 87: Eq. 1: At what level is the warming specified? For this expression to be accurate, the warming should be weighted over the atmospheric column in which the water vapour resides. This is different for Eq. 2 (l. 92), where near-surface warming can be used.

⇒ This is near-surface warming. This is an approximation that works because the troposphere is warmed relatively uniformly from the surface to ~300 hPa (Donat-Magnin et al. 2021). This will be added to the text.

l. 103: surface -> near-surface (I assume)

⇒ This will be corrected as suggested.

l. 108: " The m parameter is introduced to avoid unrealistically high melt rates" Can you provide a value, and how is it determined? Ah, it is provided in l. 120, but is given in kg m-1 s-1. Can you provide a more intuitive value, i.e. what this means per year?

⇒ We will add "$1.80 \times 10^{-4}$ kg m$^{-2}$ s$^{-1}$, i.e., 15.5 mm/day" (this is also 5.7 m/year).

l. 109-112: This is unclear, please clarify.

⇒ We will better explain this part in the revised manuscript. To reconstruct the annual melt rate (or SMB minus runoff) at a given location and on a given year, an option is to start from the annual melt rate in another simulation and to correct this value to account for the air temperature difference ($\Delta T$) between the two simulations. Given internal climate variability and uncertainty in the exponential fits, there is no reason to only use

1986 from simulation A to reconstruct 1986 in simulation B. So we reconstruct 1986 in simulation B twenty times, from years between 1976 and 1995 (and different ΔT values for every reconstruction) in simulation A, then we take the average of these 20 reconstructions.

We process similarly when we extend a simulation back in time: for example, 1950 (or any other year before 1980) is reconstructed 20 times from individual years in the corresponding MAR simulation that covers 1980–1999, and then we take the average.

This removes a part of the internal climate variability in the melt or SMB pattern of a given year (e.g., the influence of a strong synoptic event at a given year) but keeps the internal climate variability of air temperature in the reconstructed timeseries.

l. 180: typo "ooverestimated"

⇒ This will be corrected as suggested.

l.231: " Our projections over the grounded ice until 2100 agree quite well with previous estimates of sea level contribution reported by the IPCC for the three scenarios". Can these numbers be compared, i.e. did these IPCC assessments also report estimates based on SMB-only?

⇒ Yes, the IPCC emulated SMB values from the AR5 to the SSP scenarios (IPCC-AR6-WG1 Tab. 9.3). The values are already reported in our Tab. 4 which is pointed to at the end of the quoted sentence.

Section 3.2: It would be informative to provide the modelled atmospheric warming of the various models in addition to the SMB and runoff. Main motivation is that some models project negative SMB over the grounded ice sheet in the 22nd century, which must require a very strong warming, as well as significant rainfall. For these extremes, how well does the initial assumption hold that rainfall remains small?

⇒ This is a good suggestion. We will add maps of near surface warming at 2200 for the two groups of models shown in Fig. 9.

⇒ The assumption that rainfall still has a negligible role in much warmer climate is probably good given that:
  o Melt rates increase much more than rainfall rates in warmer climate (Tab. 2 of Donat-Magnin et al., 2021 and Fig. 2 of Kittel et al., 2021).
  o Melting is much more efficient than rainfall at saturating the firn with liquid water because melting removes snow/firn and converts it to liquid water while rainfall just adds liquid water (Appendix B of Donat-Magnin et al., 2021).

l. 293: lead -> led

⇒ This will be corrected as suggested.

l. 294: Georges -> George

⇒ This will be corrected as suggested.

Theoretically, it can take an almost infinite amount of time if the threshold is just passed.

$\Rightarrow$ Yes, this is exactly what we meant, we will replace "more than a decade" with "an infinite amount of time".

$\Rightarrow$ Thank you, this will be corrected.

---

## Author Response (AR1)

**Response to anonymous Referee #1.**

We thank the referee for the time spent on our manuscript and for these comments. We have realised that several aspects of our methodology were not so clearly explained and we believe that these comments will greatly improve the clarity of our manuscript. We believe that most of the points raised by the referee can be addressed by clarifying our method and we hereafter provide more details on how we have addressed these comments.

We have also made some changes that were not directly suggested by the referees but that will hopefully improve the article:

- The use of "emulation" has been generalised for the three methods instead of a mix of "extension", "reconstruction" and "emulation".
- We use the same weights for the 16-model ensemble (until 2100) and the 8-model ensemble (after 2100), which actually gives a more realistic representation of the ECS likely range than with equal weights after 2100 as implemented in the initial version, and gives more continuous results around 2100.
- We have removed the "approach" subsection that was initially presented in the results, to have all the methods described in section 2 and only the results of the ensemble of projections in section 3.

1) The distinction between "meltwater production" and "runoff" is unclear throughout the paper. This is especially the case as the authors investigate both runoff as a contribution to sea level changes and the "emergence of runoff conditions necessary for hydrofracturing". I find these two outcomes of 'runoff' a bit contradictory because if meltwater runs off into the ocean and contributes to sea level rise, then it can't induce hydrofracture events, which result from the pressure of ponded meltwater (i.e. Bell et al 2018). I think the authors should be careful to clarify this terminology throughout the paper. For example, in L27 the authors state: "The exact warming level needed to trigger important production of runoff on a given ice shelf depends on the amount of snowfall and on the snow/firn temperature and density (Donat-Magnin et al 2021, van Wessem et al., 2023)" However, these studies specifically look at meltwater production, which is the important component for hydrofracture on ice shelves, not runoff.

⇒ We acknowledge that our initial manuscript did not use a clear terminology and that some assumptions were not clearly stated. We have significantly modified the Introduction and the Methods section to make things clearer:

*"Runoff is a negative contribution to the surface mass balance. It is produced if surface melt and/or rain rates are high enough to (i) percolate and bring the temperature of underlying snow and firn layers to the freezing point, (ii) saturate the pore space in the snow and firn layers, which is sometimes referred to as firn air depletion (Pfeffer et al., 1991; Kuipers Munneke et al., 2014; Alley et al., 2018; Donat-Magnin et al., 2021), and (iii) flow into the ocean. The liquid water beyond firn saturation, hereafter referred to as "liquid water in excess", does not necessarily flow into the ocean. Liquid water in excess can indeed alternatively form ponds or be transported horizontally within the firn or at the ice surface (Kingslake et al., 2017; Bell et al., 2018).*

*In some circumstances, the presence of liquid water in excess can trigger ice shelf break-up through hydrofracturing: in favorable conditions of ice shelf stress, the weight of liquid water can destabilize a fracture and lead to its unstoppable propagation as long as liquid water keeps filling the fracture (Weertman 1973; Lai et al., 2020). Stress variations associated with surface meltwater ponding and drainage, causing flexure and fracture, can amplify this mechanism and propagate its effects spatially (Banwell et al., 2013, 2019). An entire ice shelf break-up nonetheless likely requires large amounts of meltwater production all over its surface, as observed before the break-up of*

*Larsen A in 1995 and Larsen B in 2002 ( Skvarca et al., 2004; Scambos et al., 2003; van den Broeke et al., 2005; Sergienko et al., 2005; Robel1 et al., 2019; Wille et al., 2022).*

*[…]*

*In the next section, we first use our methodology to estimate the SMB contribution to changes in sea level. As we will see in the next section, very warm conditions may lead to runoff production over the grounded ice sheet, as high as ~1000 m high, and a large part of the melt water not retained in the firn is expected to drain into ice shelves located downstream (Kingslake et al. 2017). We therefore assume that all liquid water in excess over the grounded ice sheet flows to the ice shelves downstream or directly into the ocean. Therefore, as far as the grounded ice sheet is concerned, the production of liquid water beyond firn saturation is a negative term in our SMB calculations.*

*We then use our methodology to estimate when the surface conditions have the potential to trigger ice shelf hydrofracturing. The relatively flat ice shelves are treated in a different way than the grounded ice. Indeed, in addition to the liquid water produced locally, the ice shelves receive the liquid water that was produced beyond firn saturation over the grounded ice sheet. To account for this, we assume that an ice shelf receives a fraction of the liquid water produced over the grounded ice of its drainage basin. The fraction is taken as the fraction of the basin coastline occupied by the ice shelf.*

*Another specificity of ice shelves is that they are relatively flat and can bend, so that it is impossible to estimate the amount of liquid water forming ponds or flowing into the ocean without a dedicated hydrology–firn–ice-shelf model. This is why we introduce an empirical threshold on the production rate of liquid water in excess to assess the potential for hydrofracturing. The idea is to have a rate that is sufficiently high to form ponds and fill crevasses even if a part flows into the ocean".*

We have also moved the part about the ice shelf SMB to Appendix C, which helps clarify our two main objectives: (i) SMB (including runoff) over the grounded ice sheet and sea level, and (ii) production of liquid water beyond firn saturation and related potential for ice shelf hydrofracturing.

2) Methodology
There are many parts of the methodology that are not well explained and remain unclear. This makes it challenging to assess the robustness of the methodology and therefore the results. I suggest a graphical figure detailing the methodology.

⇒ This is a very good suggestion. We have added two figures, one with a schematic for each of the three emulation methods (Fig. 1 in the revised manuscript), and one for the actual methodology used to build the ensemble of simulations from 1850 to 2100 or 2200 (Fig. 6 in the revised manuscript).

Below are some specific sentences or sections I did not understand or believe require additional detail:

a) L109: "To extend surface variables to a given local warming or cooling level, we always start from 20 different years (i.e. different values of Tref), then we average the 20 extended values." What does this mean? Does this essentially create a smoothed reconstruction?

⇒ We hope that the new schematics (Fig. 1) and a slightly reworded text in section 2.2.1 will make this part clearer:

*"In section 3, we emulate surface conditions for periods (Fig. 1a), scenarios (Fig. 1b), or CMIP models (Fig. 1c) that are not covered by existing MAR simulations. […] To emulate a*

*surface variable at a given warming or cooling level, we always calculate the emulation from 20 different years (i.e., different values of $T_{ref}$ and $\Delta T$), then we average the 20 emulated values (Fig. 1). This is done to better sample natural variability and to generate an emulated variability that is mostly related to the CMIP model temperatures. It also makes the emulation more robust from a statistical point of view".*

b) L114: "The a and b parameters are obtained through a least-mean-square-fitting of an exponential curve for SMB minus runoff on the one hand and the surface melt rate of the other hand." A supplementary figure off this exponential curve would be very helpful.

⇒ We have added this figure in Appendix B as suggested. We have also moved a part of the previous text about this fit to the Appendix, which hopefully makes the Method section easier to read.

c) L115: "The fit is done on the original model grid as regridding does not preserve exponential relationships." I find this statement to be concerning. My understanding is that the exponential relationship between the two variables may weaken in the regridding but should remain? Further, if the exponential relationship parameters are fit on the original grid, and this relationship is not preserved in the regridding, is it then appropriate to apply this fit on the regridded data?

⇒ Yes, in principle, regridding weakens rather than removes the exponential relationship, but we have removed this statement because the simulations were interpolated conservatively from a 35 km grid to a 4 km grid, so the values on the interpolated grid are almost the same as on the original grid.

d) I am unsure how to interpret the r parameter (Eq. 3, 4, L121-124). Is this the percent of excess meltwater production that is converted to runoff (as opposed to that which ponds or refreezes)? If so, I expect that this value might be different on the grounded ice sheet vs ice shelves due to due to higher slopes on the grounded ice sheet.

⇒ This has been clarified using the new formulations (see 1st comment). The *r* parameter is a threshold over which liquid water is produced beyond firn saturation, which occurs when the melt rate exceeds what can be stored and refrozen in the ongoing snow/firn accumulation. Here we do not attempt to know whether the excess of liquid water forms ponds or flows directly into the ocean (runoff), which is why the same *r* value is used on the grounded ice sheet and on ice shelves.

e) Section 2.2.4: It is unclear how you reconstruct a cooler scenario from a warmer one. Do you use SSP5 to reconstruct SSP1? Or use warmer years as a reference time and reconstruct back in time? Also, what is the purpose of this and how will this be useful?

⇒ We have added schematic (Figs. 1,6) and have reorganised the Methods section to (*i*) better show the purpose of individual emulation methods, and (*ii*) better explain how these methods are combined in the final ensemble.

To answer this specific question: if we have SSP5 over 2101-2200, we mostly use it to emulate 2101-2200 under SSP1 because this is more accurate than emulating the entire period from 2081-2100. And it is just for 2101-2120 that we combine both methods in a ramping transition.

f) L157: "Similarly as in the previous subsection, each reconstructed year is the average of 20 reconstructions from a reference ranging from 10 years before to 9 years after the reconstructed year."

I don't fully understand this sentence and have had to read it several times. It is perhaps related to the point mentioned in a) above?

⇒ Yes, see previous response. This has been clarified through the schematics and the text.

g) Section 2.2.5: A graphic or schematic outlining the workflow here would be extremely helpful because I don't understand how the emulation is done. Additionally, Figure 3 is not very intuitive for me and should be better explained as I am unsure how to interpret it.

⇒ Yes, the workflow is now outlined in Fig. 6.

We have rewritten the caption of Fig. 3 (now Fig. 4) to make it easier to understand. This way of presenting the results is also found in other papers (e.g., Figs. 4 & 7 in Barthel et al., 2020).

h) Figure 4: In panels e and f, the reconstructed runoff anomaly is too low, despite the melt anomaly (panel c and d) being fairly accurate for ice shelves and too high over the grounded ice sheet. Does this suggest some mis-parameterization in your method? Perhaps the wrong r value?

⇒ The reconstructed runoff anomalies are both slightly too high in Fig. 4 (now Fig. 5), as indicated by the positive bias values indicated in brackets for panels c, d, e, f. If the referee was thinking about Fig. 2 (now Fig. 3) and not Fig. 4, there is indeed a positive bias in the melt anomaly and a negative bias in the runoff (i.e., excess of liquid water beyond firn saturation). Decreasing the *r* value would clearly reduce the runoff bias in this figure, but not in others. This indeed shows that the parameterisation is not perfect. We have added a comment on this.

i) L205-208: For these seven simulations… apply a ramping transition between the two methods from 2101 to 2120." I don't follow what is being done here and again, I think a figure or something would help the reader understand the methodology.

⇒ Yes, this has been better explained thanks to the addition of a schematic (Fig. 6), and the corresponding text has been rewritten.

j) L280: The choice of a 100kg/m2/yr runoff threshold for triggering hydrofracture seems extreme and is not well-defended in the text. This is 50-67% less than the average meltwater production estimated prior to the collapse of Larsen B (200-300 according to the text). How was this threshold "empirically" chosen? Was it just based on Larsen A/B?

⇒ We have improved this part of our manuscript by using a threshold distribution to better account for its uncertainty:

*"Another specificity of ice shelves is that they are relatively flat and can bend, so that it is impossible to estimate the amount of liquid water forming ponds or flowing into the ocean without a dedicated hydrology–firn–ice-shelf model. This is why we introduce an empirical threshold on the production rate of liquid water in excess to assess the potential for hydrofracturing. The idea is to have a rate that is sufficiently high to form ponds and fill crevasses even if a part flows into the ocean. The average production of liquid water beyond saturation over Larsen B prior to its collapse was estimated between 200 and 300 kg $m^{-2}$ $yr^{-1}$ (Holland et al. 2011; van Wessem et al., 2016; Costi et al., 2018), so our threshold has to be lower than that. There is nonetheless a large uncertainty on the threshold and we sample it in a normal distribution of 150 and 61 kg $m^{-2}$ $yr^{-1}$ of mean and standard deviation, respectively. This is chosen to obtain 90% of the threshold values between 50 and 250 kg $m^{-2}$ $yr^{-1}$. The lower*

*end of this range is chosen empirically so that not too many ice shelves are above the threshold in present-day conditions. The uncertainty on the threshold is hence included in the calculation of the probability of a given ice shelf to be over the threshold".*

k) L291: How do you define "likely" or "very likely"?

⇒ This was indeed only defined in some table and figure captions. We have added the definition in the main text: "*we present our results as confidence intervals, which we define as in the IPCC reports: 17–83th percentiles for the likely range (66% probability) and 5–95th percentiles for the very-likely range (90% probability)*".

3) Figure 1
L148: It seems a bit of a stretch to say that the extension of the RCM simulation is suitable for 25 years over the grounded ice sheet…. Really, there is just one year anomalously high SMB year at ~2125. Otherwise, the original MAR simulation and the reconstruction have opposite trends, even during this first 25 years.
In general, Figure 1 is concerning for me. It seems that this reconstruction method cannot be applied in a warming climate. However, the authors do apply some sort of reconstruction to obtain the results in Figures 6-12. How were these reconstructions obtained when Figure 1 demonstrates issues for applying this method in a warming climate? How can we trust the results presented here in light of Figure 1?

⇒ First of all, to build our ensemble of projections, the emulation from another period is only used for the extension to 1850, i.e., to a colder climate, and in a 20-year ramp-down transition from 2101 to 2020 (towards a warmer climate). We believe that this is now much clearer in the revised manuscript, in particular thanks to the new schematics (Fig. 6).

Second, we have modified Fig. 1 (now Fig. 2) to evaluate both a backward and a forward emulation, and we have indicated the biases over 20 years. This confirms that the emulation backward (towards a colder climate) is less biased than the emulation forward (towards a warmer climate). The emulation forward nonetheless produces biases that remain small over the first 10 years, and reasonable over the first 20 years for $r = 0.5$ and $r = 0.6$ (~10% bias for the SMB over the grounded ice sheet and the production of liquid water in excess over the ice shelves, Fig. 2a,f). We also want to point out that our intention is not to assess the detail of the emulated interannual variability, but to describe the overall bias that could affect an ice-sheet model in the long term.

4) Relation to previous studies
In general, this manuscript is lacking some references to and context within recent literature. For example, how do the results in section 3.3 add to or fit within the context of previous ice-shelf potential instability studies (i.e. van Wessem et al., 2023; Dunmire et al., 2024; Alley et al., 2018; Lai et al., 2020)).

Additionally, some reverences to previous AIS SMB studies are missing (e.g., Gorte et al., 2020, Noel et al 2023).

⇒ We have added more comparisons to similar types of predictions. Some studies were published just before (Noel et al., 2023) or just after (Dunmire et al., 2024) our initial submission, but we can now include them. We have also included the recent projections by Veldhuijsen et al. (2024). All these references are now cited in our new section 3.2

(previously 3.3) together with references that were already cited (Kuipers Munneke et al., 2014; Donat-Magnin et al., 2021; van Wessem et al., 2023).

Regarding the other articles mentioned by the reviewer, Lai et al. (2020) was already mentioned as describing the mechanical conditions that can lead to hydrofracturing in the presence of meltwater beyond firn saturation. We have cited Gorte et al. (2020) for their reduction of uncertainty by weighting CMIP model based on their misfit with observations and Noel et al. (2023) for their novel downscaling method, but we do not compare our AIS SMB estimates to theirs because Gorte et al. do not account for runoff and because the projection of Noel et al. is based on a single CMIP model.

We have also added a paragraph in the Conclusion to explain what our projections bring compared to the aforementioned studies:

"*We believe that most original aspects of our projections compared to recent studies (van Wessem et al., 2023; Dunmire et al., 2024; Veldhuijsen et al., 2024) are (i) the time coverage back to 1850 and until 2200 while other studies cover ~1980-2100, (ii) the use of a large weighted ensemble of CMIP models to account for the models uncertainty while keeping a plausible equilibrium climate sensitivity, (iii) the relatively large number of MAR simulations used to assess and calibrate our simple emulator*".

5) Finally, the motivation for this work, and specifically how this method could be used in the context of ISMIP7, should be elaborated.

⇒ We have reworded a paragraph of our conclusion:

"*Our method is useful to populate ensembles of surface mass balance and production rates of liquid water which are needed to constrain ice sheet model ensembles and to estimate the likely range of future sea level rise. This approach has been used to complete the set of RCM simulations used to drive ice sheet simulations until 2150 in the PROTECT European project (Durand et al., 2022; Mosbeux et al., 2024). This could also be useful for the upcoming ISMIP exercise as it is relatively simple to use, it can be applied before obtaining all the CMIP 6-hourly data and before running RCMs, and it can provide the information needed to trigger the hydrofracturing mechanism in ice sheet models*".

Minor comments

L7: "After correcting the distribution of equilibrium climate sensitivity of 16 climate models…" From just the abstract, it is unclear what this means.

⇒ We have reformulated as "*After weighting 16 climate models to obtain a realistic distribution of the equilibrium climate sensitivity, we find…*". We believe that the concept of "equilibrium climate sensitivity" is sufficiently known in the climate community to appear in the abstract, but we have provided a definition in section 2.2: "*the global mean surface air temperature increase that follows a doubling of atmospheric carbon dioxide*".

L7: "… we find a likely contribution of surface mass balance to sea level rise of 0.4 to 2.2 cm from 1900 to 2010…". It does not make sense that the contribution of SMB to SLR would be positive for this period so I'm assuming this is with respect to a reference period? Same for the SLR contribution ranges in the following lines?

$\Rightarrow$ Embarrassingly, the entire sentence was wrong and has been replaced with:

*"we find a likely contribution of surface mass balance to sea level rise of **-2.2 to -0.4** cm from 1900 to 2010, and -3.4 to -0.1 cm from **2000** to 2099 under the SSP1-2.6 scenario, versus -4.4 to -1.4 cm under SSP2-4.5 and -7.8 to -4.0 cm under SSP5-8.5"*.

L25: "Hydrofracturing may strongly enhance the contribution of upstream glaciers to sea level rise." This is a bit misleading. The papers cited (among other work) indicate that the removal/collapse of ice-shelves (perhaps due to hydrofracture events) causes a speed-up of upstream glaciers, not just the hydrofracture event itself.

$\Rightarrow$ We have reworded as:

*"When occurring on ice shelf parts that buttress the upstream flow, hydrofracturing and the resulting ice shelf collapse may strongly enhance the contribution of upstream glaciers to sea level rise"*.

L41: "Because of these difficulties, only… which is generally insufficient to sample the CMIP model diversity." The "- when produced –" in this sentence threw me off a bit and I had to read it a few times to understand what was being said.

$\Rightarrow$ Agreed, we have removed this part of the sentence.

L43: "… correct unrealistic Equilibrium Climate Sensitivity…" A brief explanation for this concept would be helpful here.

$\Rightarrow$ We have removed the mention to the concept of Equilibrium Climate Sensitivity from the Introduction, and we now define it in section 2.2.

L45: "Over the years, Antarctic Ice Sheet modellers have often scaled their best estimates of present-day accumulation to temperature anomalies from the CMIP models…". This sentence fragment is unclear to me.

$\Rightarrow$ We have replaced with:

*"Over the years, Antarctic Ice Sheet modellers have often scaled their best estimates of present-day accumulation to temperature anomalies from the CMIP models (e.g., based on the Clausius-Clapeyron relationship as in Gregory and Huybrechts, 2006)"*.

L67: "The surface mass balance and melting… Donat-Magnin et al (2020) and Kittel et al (2021)." I think a brief explanation of the results of these papers would be helpful here. I am left wondering: And how does MAR do in comparison to observational products?

$\Rightarrow$ In the new Appendix A, we have summarised the main results of the melt and SMB evaluations presented in previous articles and we have added a plot to further evaluate MAR's runoff in comparison to a satellite estimate of melt pond volume.

L101: Assuming that all precipitation is entirely made of snow is a big assumption to make, especially for projections that extend to 2200 in high-emission scenarios. The impact of this assumption should at least be discussed somewhere in the paper.

$\Rightarrow$ We already had this sentence:

*"The presence of rainfall makes Eq. (3) more complex (see Appendix B of Donat-magnin et al. 2021), but rainfall is generally negligible compared to snowfall in present-day conditions and to surface melt rates in much warmer conditions (Donat-Magnin et al., 2021)".*

We have added:

*"Furthermore, the only RCM simulation until 2200 (MAR–IPSL-CM6A-LR, SSP5-8.5) does simulate the actual rainfall/snowfall distribution and still has a positive surface mass balance over ice shelves in 2200 (see section 3), which indicates that snowfall is still dominant over rainfall".*

L132-134: Should this really be interpreted as a 'mass loss rate' if Figure 1 shows anomaly values with respect to a reference period? The SMB for the reference period is positive (although the specific reference period value from MAR should be mentioned somewhere in the paper). Even though the line in Figure 1b decreases throughout the timeseries, mass loss doesn't occur until it reaches the negative magnitude of the reference period. For example, if the reference period SMB for ice-shelves ~500 Gt/yr, then surface mass loss doesn't really occur until approximately 2120 (when the time series reaches -500 Gt/yr).

⇒ Yes, this has been reformulated as:

*"the SMB in the original MAR simulation remains steady until ~2090 then drops due to increased melting with a SMB in 2200 that is 2000 Gt yr⁻¹ lower than in 1995–2014".*

L138: "… although this is still an improvement compared to the original IPSL-CM6A-LR outputs". I think it would be very interesting to have these original ESM timeseries plotted in Figure 1 as well.

⇒ We have added the SMB time series directly calculated from IPSL-CM6A-LR. See the black dashed lines in new Fig. 2a,b.

L123: "… covering the aforementioned range, i.e. 0.5 to 0.9." This range is different from that mentioned before (0.6-0.85, L98).

⇒ We have reformulated as:

*"we will assess values in the 0.50–0.90 range, which includes the values used in previous work".*

Section 2.2.3: Somewhere in this section it should be clarified that Equation 4 is used to do this reconstruction.

⇒ A reference to Eq. 4 has been added.

L179-181: "The realistic SMB reconstructions derived from MAR-ACCESS1.3 are mostly compensations between overestimated melt and overestimated accumulation". Why do you say this?

⇒ Because the melt rates emulated from MAR-ACCESS1.3 are largely overestimated, as seen by the large green pentagon in Fig. 3c,d, which means that the realistic SMB is due to underestimated accumulation. This is to argue that emulations from MAR-ACCESS1.3 are outliers despite realistic emulated SMB over the grounded ice sheet. A reference to Fig. 3c,d has been added to make this sentence clearer.

L210: "… with a 20-year transition." What does this mean?

⇒ This has hopefully been clarified, see previous responses.

L238: Figure 8 is mentioned before Figure 7

⇒ This has been corrected.

L253: What does "weaker SMB" mean?

⇒ This has been replaced with "lower SMB".

L265: "Spatially, a net surface mass loss arises for several ice shelves…" I find this to be a bit misleading since Figure 7 shows SMB anomalies from a reference period, not absolute SMB. Is there actually a net surface mass loss or is it just lower than the reference period?

⇒ Agreed, this has been reformulated.

L294 – It should be mentioned that George VI ice shelf has compressive stresses which do not promote hydrofracture occurrence (Labarbera et al., 2011)

⇒ We have added this statement and the suggested reference.

L 311: What is the A1B scenario?

⇒ This is a scenario that was used in CMIP3 and IPCC-AR4. This entire paragraph has been rewritten with no mention of the A1B scenario.

Technical corrections:

⇒ These have been corrected.

L33: progresses à progress
L62: Citation needed for the pore close-off density.
L180: "ooverestimated"
L185: "emulation" à emulations
L234: "scenario" à "scenarios"
L 254: End parenthesis after "(Fig. 9."
L262: "to the exception" à "with the exception"
Figure 8 caption: "same as Fig. ?? is also shown"

**Response to anonymous Referee #2.**

We thank the referee for the time spent on our manuscript and for these comments. We have realised that several aspects of our methodology were not so clearly explained and we believe that these comments will greatly improve the clarity of our manuscript. We believe that most of the points raised by the referee can be addressed by clarifying our method and we hereafter provide more details on how we have addressed these comments.

We have also made some changes that were not directly suggested by the referees but that will hopefully improve the article:
- The use of "emulation" has been generalised for the three methods instead of a mix of "extension", "reconstruction" and "emulation".
- We use the same weights for the 16-model ensemble (until 2100) and the 8-model ensemble (after 2100), which actually gives a more realistic representation of the ECS likely range than with equal weights after 2100 as implemented in the initial version, and gives more continuous results around 2100.

We have removed the "approach" subsection that was initially presented in the results, to have all the methods described in section 2 and only the results of the ensemble of projections in section 3.

General
This is an interesting paper that addresses an important problem: when can we expect runoff from Antarctic ice shelves, leading to mass loss and/or meltwater ponding which is commonly regarded as a precursor for ice shelf hydrofracture. The authors use multiple MAR simulations of future Antarctic climates, with which they calibrate a simple statistical model to emulate melt, runoff so as to approximate SMB and (instantaneous) firn saturation. Not considered are sublimation, rain and the time it takes to saturate the firn. When compared directly to MAR output, the emulator gives mixed results, yet provides valuable insights in the uncertainties that arise from the driving global models. The writing can be clarified in places (see below), the figures are generally clear.

**Major comments**

The abstract is at places confusing. It goes in one step from describing runoff emulation to AIS SLR contribution. Although it is stated that the mass loss numbers are based on SMB alone, the link that is made to hydrofracturing in the title and the presented numbers in terms of sea level rise could trick the reader into thinking that dynamical effects are also considered. It should be made clear from the outset is that this study treats runoff in two ways: as a source of mass loss and as a threshold to induce ponding and hydrofracturing (after which the dynamical effect on the ice sheet is not considered).

$\Rightarrow$ We have reworded the abstract to hopefully make our approach clearer. The introduction and methods have also been reworded and completed along these lines.

Also in the abstract: "Based on a more limited and uncorrected ensemble, we find a considerable uncertainty in the contribution to sea level from 2000 to 2200". It is unclear whether this enhanced uncertainty derives from the fact that the ensemble is smaller and uncorrected, or from the method, or all of these. Please only include major results in the abstract, the details can be elaborated upon in the text.

$\Rightarrow$ The sentence has been rewritten as:

*"The contribution from 2000 to 2200 is highly uncertain: between -10 and -1~cm in SSP1-2.6 and between -33 and +6~cm in SSP5-8.5 depending on the model".*

l.59: The statistical model is based on MAR data, which is a fair choice given that this model has been used for multiple future runs. But for the statistical model to be useful, MAR must provide reliable runoff. Here it is stated that the atmosphere in MAR is coupled to a 30 layer snow model "...representing the first 20 m of snow/firn with refined resolution at the surface". But in many places the firn layer in Antarctica is considerably deeper than 20 m? What does this imply for runoff simulated by MAR? See also my comments on treatment of percolation and runoff evaluation below.

⇒ Only using MAR is certainly a caveat but this was the only RCM for which we had an ensemble of projections sufficiently large to be robustly emulated. This caveat was already acknowledged in the Discussion section. We nonetheless agree that simulating 100 m of firn would be better than 20 m, although the potential presence of ice slabs makes it difficult to anticipate the exact effect. The firn depth is expected to change the timing of firn saturation/air depletion, but not the threshold for firn saturation which is more related to the melting and snowfall rates. We have added the sentence in red to the relevant paragraph in the Discussion section:

"*Our approach has consisted of emulating MAR simulations. Other RCMs, possibly combined to elaborated firn models, have similar skills in representing typical Antarctic conditions (Mottram et al., 2021), but there is likely a considerable spread in their response to surface warming. For example, the depth and vertical resolution of firn models probably make important differences in the timing of runoff production. The 20-m firn layer simulated in MAR thus likely reaches liquid water saturation earlier than models with a thicker firn layer. One of the next priorities will therefore be to emulate the diversity of RCM sensitivities, which would make the uncertainty ranges much more comprehensive.*"

l.65: " If liquid water is not able to percolate further down, then it fills the entire porosity space of surface layers, and the excess is considered as runoff and removed from the snow/firn model (there is no representation of ponds or horizontal routing)". From this description it is not clear to me how water percolation interacts with ice lenses in the firn layer and how the process of filling up works before runoff starts.

⇒ This sentence has been clarified so that the entire paragraph reads:

"*In the presence of surface melting or rainfall, liquid water percolates downward into the next firn layers with a water retention of 5% of the porosity in each successive layer. The firn layers are fully permeable until they reach a close-off density of 830 kg m$^{-3}$. To account for possible cracks in ice lenses and moulins, the part of available water that is transmitted downward to the next layer decreases as a linear function of firn density, from 100% transmitted at the close-off density to zero at 900 kg m$^{-3}$ and beyond. If liquid water is not able to percolate further down, it remains where it is. When the entire porosity space in the surface grid cell is filled with liquid water or if the surface grid cell is denser than 900 kg m$^{-3}$, any additional surface melt is considered as runoff and removed from the snow/firn model. There is no representation of ponds or horizontal routing.*"

l.67: "The surface mass balance and melting conditions produced by MAR have been evaluated in comparison to observational." Has there also been an attempt to evaluate the modelled meltwater ponding/runoff? I seem to remember that Van Wessem et al. (2023) compared their predicted ponding potential to satellite observations of melt ponds.

⇒ First of all, we have included more details on the comparisons to observations in this paragraph following a comment by Reviewer #1. Van Wessem et al. (2023) indeed presented a comparison of their RACMO simulations to an observational estimate of

melt pond volume (derived from Sentinel 2). This comparison is qualitative as neither MAR nor RACMO simulate ponding (they remove the excess of liquid water and do not simulate horizontal transport of liquid water). We nonetheless show here the runoff produced by MAR forced by ERA5 over 2015-2022 in comparison the melt pond volume over the same period:

[Figure]

We find that the areas of high runoff in MAR generally correspond to the areas where high melt pond volumes are estimated from Sentinel 2 data, even if the area of high runoff over Larsen C is larger than the area of large melt pond volume in the satellite product. This comparison has been presented together with other results from published studies in Appendix A.

Figure 1 shows results for the integrated grounded ice/ice shelves only. How do the spatial patterns look?

⇒ We have added three figures in Appendiix C, one for each type of application of our evaluation method. The patterns are well reproduced.

**Minor comments**

l. 12: "Based on a runoff criteria.." Please note that 'criteria' is plural, so either use "Based on runoff criteria.." or "Based on a runoff criterion.."

⇒ This has been corrected.

l. 12: "we identify the emergence of surface conditions prone to hydrofracturing" Suggest: "we identify the timing of surface conditions that make ice shelves prone to hydrofracturing" or something similar.

⇒ This has been changed.

l. 13: " A majority of ice shelves could remain safe" Suggest to reformulate: "Our results suggest that the majority..."

⇒ This has been modified as suggested.

l.16: precipitation -> snowfall

⇒ Precipitation consists of snowfall + rainfall, and rainfall is a positive term in the surface mass balance that effectively contributes to mass gain until the firn is saturated with liquid water or until the surface consists of bare ice. But rainfall has a minor effect and this first sentence may be clearer with "snowfall", so we have corrected as suggested.

l. 18: However, if air temperatures exceed -> However, model simulations suggest that, if atmospheric warming exceeds

⇒ This has been modified as suggested.

l. 22: can be conducive of -> can be conducive to

⇒ This sentence has disappeared from the revised manuscript.

l. 31: sea level -> sea level change

⇒ This has been modified as suggested.

l. 36: in particular with regard to firn saturation by meltwater and subsequent runoff -> in particular with regard to firn saturation by meltwater and subsequent ponding

⇒ This has been modified as suggested.

l. 87: Eq. 1: At what level is the warming specified? For this expression to be accurate, the warming should be weighted over the atmospheric column in which the water vapour resides. This is different for Eq. 2 (l. 92), where near-surface warming can be used.

⇒ We have added this paragraph to clarify this point:

*"Hereafter, we assume that $\Delta T$ is a variation in near surface air temperature in both Eq. 1 and Eq. 2, which is a reasonable approximation given that the troposphere warms relatively uniformly from the surface to ~300 hPa (Donat-Magnin et al., 2021, their Fig. 1)".*

l. 103: surface -> near-surface (I assume)

⇒ This has been modified as suggested.

l. 108: " The m parameter is introduced to avoid unrealistically high melt rates" Can you provide a value, and how is it determined? Ah, it is provided in l. 120, but is given in kg m-1 s-1. Can you provide a more intuitive value, i.e. what this means per year?

⇒ We have added "$1.80 \times 10^{-4}$ kg m$^{-2}$ s$^{-1}$, i.e., 15.5 mm/day" (this is also 5.7 m/year).

l. 109-112: This is unclear, please clarify.

We have modified the text and added schematics in two figures (now Fig. 1 and Fig. 6) to better explain our method.

l. 180: typo "ooverestimated"

⇒ This has been corrected.

l.231: " Our projections over the grounded ice until 2100 agree quite well with previous estimates of sea level contribution reported by the IPCC for the three scenarios". Can these numbers be compared, i.e. did these IPCC assessments also report estimates based on SMB-only?

$\Rightarrow$ Yes, the IPCC emulated SMB values from the AR5 to the SSP scenarios (IPCC-AR6-WG1 Tab. 9.3). The values were already reported in our initial Tab. 4 (now Tab. 3) which was referred to at the end of the quoted sentence.

Section 3.2: It would be informative to provide the modelled atmospheric warming of the various models in addition to the SMB and runoff. Main motivation is that some models project negative SMB over the grounded ice sheet in the 22nd century, which must require a very strong warming, as well as significant rainfall. For these extremes, how well does the initial assumption hold that rainfall remains small?

$\Rightarrow$ We have added maps of near surface warming in 2181–2200 for the two groups of models shown in what is now Fig. 11.

$\Rightarrow$ The assumption that rainfall still has a negligible role in much warmer climate is probably good given that:
- o Melt rates increase much more than rainfall rates in warmer climate (Tab. 2 of Donat-Magnin et al., 2021 and Fig. 2 of Kittel et al., 2021).
- o Melting is much more efficient than rainfall at saturating the firn with liquid water because melting removes snow/firn and converts it to liquid water while rainfall just adds liquid water without removing firn (Appendix B of Donat-Magnin et al., 2021).
- o Furthermore, the only RCM simulation until 2200 (MAR–IPSL-CM6A-LR, SSP5-8.5) does simulate the actual rainfall/snowfall distribution and still has a positive surface mass balance over the ice shelves in 2200 (see section 3), which indicates that snowfall is still dominant over rainfall.

All this is mentioned in the revised manuscript.

l. 293: lead -> led

$\Rightarrow$ This has been corrected.

l. 294: Georges -> George

$\Rightarrow$ This has been corrected.

l. 323: "...while it can take more than a decade to saturate it if melt rates are just above the threshold..." Theoretically, it can take an almost infinite amount of time if the threshold is just passed.

$\Rightarrow$ Yes, this is exactly what we meant. We have replaced "more than a decade" with "an infinite amount of time".

l. 363: criteria -> criterion

$\Rightarrow$ This has been corrected.

---

## Author Response (AR2)

**Reviewer #2**

**General**

Compared to the original version, the paper is completely overhauled. Much of the unclarities have been resolved, and the readability of the paper has much improved, although the amount of information can still be overwhelming to the unprepared reader. Although one could still argue about the robustness of some of the results, I feel it is a valuable contribution, especially for the ice sheet modelling community in search of an easy way to implement a first-order estimate of snowfall, melt and firn saturation in their offline Antarctic ice sheet models.

⇒ We sincerely thank the reviewer for this second careful examination of our manuscript. We have revised our manuscript following most of the suggestions.

**Remaining not so minor comments**

(line numbers refer to document including tracked changes)

Paragraph starting line 29: This is a useful summary of the conditions that must be met before runoff occurs. As it is presented now, it can also be read that ANY of these conditions must be met rather that ALL. Can you make this a little more specific?

⇒ Yes, this sentence has been rewritten as:

*Runoff is a negative contribution to the surface mass balance. It is produced if surface melt and/or rain rates are high enough to **successively** (i) percolate and bring the temperature of underlying snow and firn layers to the freezing point, (ii) saturate the pore space in the snow and firn layers, which is sometimes referred to as firn air depletion (Pfeffer et al. 1991, Kuipers Munneke et al. 2014, Alley et al. 2018), and (iii) flow into the ocean.*

Have you considered using the simpler phrase "excess liquid water" rather than "liquid water in excess"?

⇒ Thank you for this suggestion, we have used "excess liquid water" in the revised manuscript.

l. 115: It remains unclear whether the MAR simulations are fully transient, and how/over which period the model snow/firn layer was initialized with regards to e.g. density and liquid water content. Please provide this information and how it could influence the results. Along the same lines: how should we see these results in the light of this recently published paper comparing RCMs over the Greenland ice sheet: https://agupubs.onlinelibrary.wiley.com/doi/10.1029/2024GL111902? Consider including a remark or two about this in your discussion following line 520.

⇒ We have added the following paragraph to clarify this point:

*For computational reasons, all simulation years were run in parallel with 20 years of spin-up to equilibrate the firn properties (e.g., 2051 is spun up from the transient 2031–2050 period). The initial state (e.g., 01-JAN-2031 for the simulation of 2051) is taken from the MAR–ACCESS-1.3 RCP8.5 simulation (Tab. 2), itself spun up from a previous version of MAR at 50 km resolution driven by NorESM1-M under RCP8.5 and spun up for 30 years from a present-day*

*MAR simulation. A spin up of 20 years is generally sufficient to remove any sensitivity to the initial state (Donat-Magnin et al., 2021, their Fig. 12).*

About the suggestion to add a reference to Glaude et al. (2024), there was some specific tuning for the Greenlandic MAR configuration that may explain the strong future surface mass loss compared to the RACMO model. This tuning has not been used in our Antarctic configuration, so we do not think it is relevant to go too far in the reference to Glaude et al. (2024). We have nonetheless added a reference to this paper in the discussion section to support the extension to other RCMs.

Eq. 4: I find the use of RU as a negative number confusing. If runoff is positive, it represents a mass loss, so SMB = SF - RU is a more logical convention.

$\Rightarrow$ We think this is a matter of taste and we prefer keeping RU as a negative number so that removing the runoff contribution from the SMB can be written as SMB – RU (1$^{st}$ line of eq. 4).

l.494: I was surprised to read this so I looked it up but I think that Van Wessem and others (2023) do not claim the Ross ice shelf being susceptible to hydrofracturing. In their study, the near surface air temperature in some CMIP models and scenarios rather passes the warming threshold allowing initiation of conditions that in time can lead to meltwater ponding (note that saturating the firn layer can still take a lot of time especially if melt is weak). See also your own remark about this delay in l. 514.

$\Rightarrow$ We agree. Only half of the models in van Wessem et al. (2023) are prone to ponding over the Ross Ice Shelf under SSP5-8.5, which is not sufficient to reach our likely range (defined as 66% probability). We have therefore included van Wessem et al. in the previous citations:

*The giant Ross and Ronne-Filchner ice shelves are unlikely to experience hydrofracturing before the early 22$^{nd}$ century in SSP5-8.5, consistently with previous studies (Kuipers Munneke et al. 2014, **van Wessem et al. 2023**, Dunmire et al. 2024, Veldhuijsen et al. 2024).*

**Remaining minor comments**
(line numbers refer to document including tracked changes)

Some references ran from the page in my pdf.

$\Rightarrow$ This is an issue of the track-change version, the version with no tracked changes is ok. We didn't find an easy way to fix this using the *LaTeX/Overleaf* editor and the *latexdiff* command line.

l. 62: "Because of these difficulties, only a limited number of RCM-based projections are usually produced, which is generally insufficient to sample the CMIP model diversity regarding their representation of the recent period (Barthel et al., 2020) and their sensitivity to increasing anthropogenic emissions (e.g. Hausfather et al., 2022)." This sentence is formulated a little awkwardly; consider breaking up and/or reformulating.

$\Rightarrow$ This has been rewritten as:

*Because of these difficulties, only a limited number of RCM-based projections are usually produced, which is generally insufficient to sample the CMIP model diversity. **This may affect the** representation of the recent period (Barthel et al., 2020) and **the** sensitivity to increasing anthropogenic emissions (e.g. Hausfather et al., 2022) **in the small RCM ensemble**.*

l. 70: later -> latter
⇒ This has been corrected.

l. 91: consider removing "with each other".
⇒ This has been modified as suggested.

Caption Table 1: SSP-5.85 -> SSP-5-8.5
⇒ This has been corrected.

Caption Table 1: for which at least a MAR -> for which at least one MAR
⇒ This has been corrected.

Tables 1 & 2: please try to diminish the overlapping information in these tables.
⇒ Two models in Tab. 2 are from the CMIP5 era and are not represented in Tab. 1. We have therefore kept the ECS and ensemble members in Tab. 2 even if the information was already provided in Tab. 1 for the CMIP6 models.

l. 148: "here we do not attempt to know" please reformulate.
⇒ This has been rewritten as "we do not attempt to determine".

l. 154: Is this really proof of a positive mass balance?
⇒ This paragraph has been rewritten following a suggestion of Reviewer #3 (see response to their 3rd comment).

l. 156: contribution > contributions
⇒ This has been corrected.

Caption Fig. 1: non -> not (3x)
⇒ This has been corrected.

l. 325: surface air -> near-surface air
⇒ This has been modified as suggested (also in 5 other sentences).

l. 334: There appears to be a parenthesis missing
⇒ We did not find any missing parenthesis around L. 334.

l. 353: as high as ~1000 m high -> up to elevations of 1000 m asl
⇒ This has been modified as suggested.

l. 345: into -> onto
⇒ This has been corrected (L. 354).

l. 405: over -> above
> ⇒ This has been corrected.

l. 406: a SMB -> an SMB
> ⇒ This has been corrected (also in two other sentences).

l. 487: monitored -> considered
> ⇒ This has been corrected.
* * *
**Reviewer #3**

**General comments**

This study uses a mixed physical/statistical method to dramatically increase the number of model ensemble members available to project future Antarctic SMB changes, and consequently sea level rise contributions and ice shelf hydrofracturing risk. I find it to be both novel and interesting, and feel it will be a valuable contribution to the scientific literature. The manuscript has been revised significantly in light of two reviewers' comments, and the result is a much clearer and more instructive paper. The schematics were particularly helpful to me to understand the method. The authors have addressed many of the earlier concerns of the reviewers, however I feel the paper would still benefit from some minor revisions, as detailed below.
> ⇒ We welcome this new evaluation of our work, and we thank Reviewer #3 for these very useful suggestions.

**Line numbers are in reference to the tracked changes document**

L49-52 Because you say here that the timing of ice shelf collapses influences sea level rise, it could be helpful to explicitly mention somewhere that you do not include the contribution of these collapses to sea level rise, because you don't have a dynamical ice sheet in your simulations.
> ⇒ We have added the following sentence in the introduction paragraph of section 3:
>
> *Importantly, our estimates of sea level projections only contain the part related to SMB variations, and not the contribution from the ice sheet dynamics which is driven by ocean-induced melting and hydrofracturing.*

Tab 1 caption "2300" ◊ "2200"
> ⇒ This has been corrected.

L152-155 Can you offer any quantification of how much snowfall >> rainfall in the present or melt >> rain in the future? For instance, what percentage of MAR's precip falls as rain in the historical simulation, and how much does this increase by the end of the century?

L155-157 Relatedly, can you estimate the uncertainty associated with the assumption that drifting snow and sublimation are negligible? Maybe you can put a number on the mass change associated with sublimation (either from Agosta et al. or your own MAR simulation)?

⇒ We have rewritten the corresponding paragraph in section 2.3.1 as:

*In Eq. 3, the effect of rainfall, sublimation and drifting-snow erosion are assumed to be negligible. Sublimation remains below 10% of snowfall even in a warmer climate (Kittel et al., 2021, their Tabs. S2-S3), and drifting-snow erosion is at least an order of magnitude smaller than sublimation (Gadde and van de Berg, 2024). As shown in Appendix B, rainfall represents less than 15% of the total precipitation on the grounded ice sheet until 2200 and on the ice shelves until 2100. The impact of neglecting rainfall in our method is discussed in Appendix B.*

We have added a new appendix (now Appendix B) that shows the snowfall, rainfall and melting time series for both the grounded ice sheet and the ice shelves in the IPSL–SSP5-8.5 simulation until 2200 (now Fig. B1). In this Appendix, we provide arguments supporting that rainfall can be neglected in our approach.

Fig 4 I agree with R1 that this figure and its description is slightly confusing (there's a lot of information in there!) but your edits have improved the clarity of the caption. I wonder if you could increase the clarity of the text by making explicit reference to each of the properties of the figure that lead you to certain conclusions, e.g. "First of all, the minimal differences between the black dots on each radial and the coloured dot corresponding to the same GCM show that the biases are small for MAR simulations derived from themselves (Fig. 4). This shows that our methodology and its implementation are robust."

⇒ We have followed this suggestion and this paragraph now starts as:

*The method is evaluated in Fig. 4. First of all, the minimal differences between the black dots on each radial and the coloured dot corresponding to the same climate model show that the biases are small for MAR simulations derived from themselves. This shows that our methodology and its implementation are robust. We nonetheless note larger differences between the black dots on each radial and the coloured dots corresponding to the other climate models in Fig. 4c,d, indicating significant biases in melt rates when a MAR simulation is derived from another one.*

L260-262 It doesn't follow so well that you talk about the biggest limitation of this part of the method (that runoff is underestimated at the end of the 21st century in the ssp3-4.5 emulated scenario) and then immediately say that you conclude the method is adequate. Another statement to qualify this would help the reader understand your conclusion, particularly as it makes the end of century emulated SMB a different sign to the original MAR simulation. e.g. can you say something about the relative magnitude of the underestimate in runoff, or say something about the performance of the emulated simulation over the course of the entire period?

⇒ We agree. This paragraph has been rewritten as:

*The SSP1-2.6 and SSP2-4.5 emulated fields are quite accurate over the grounded ice sheet during the 21st century: the biases indicated in Fig. 3 for the emulated values are relatively low compared to the mean values simulated by MAR. The bias in emulated runoff becomes larger at the end of the 21st century in the SSP2-4.5 scenario (Fig. 3e-f). This bias has little impact on the grounded ice sheet SMB (Fig. 3a) but cancels the small negative SMB anomaly simulated*

*over the ice shelves near 2100 (Fig. 3b). These biases are small and limited to the end of the 21st century, so we conclude that our method is suitable for the emulation of multiple SSP scenarios based on an existing MAR simulation in a warmer scenario (here SSP5-8.5). The spatial patterns are also well represented by this method (Appendix D, Fig. D2).*

L297-298 I think this statement could be clearer. You could explicitly state that changing the r value would not be consistent with the results of the previous sections, nor reflect physical processes.

⇒ This has been modified as suggested.

L303 Given that averaging across CMIP-model emulations is required to achieve a useful result (which makes sense given the uncertainty associated with clouds and other parameterisations in GCMs), can you offer guidance on the way these results and/or the method should be interpreted by others wishing to replicate your approach?

⇒ We have added the following statement in the Discussion section:

*For any method, the results presented in section 2.3.5 stress the importance of using a training dataset made of RCM simulations driven by multiple CMIP models for obtaining robust results.*

Figs 1, 6 These schematics are really helpful for aiding understanding!

⇒ Thank you!

L377 should this be "17th-83rd"?

⇒ This has been corrected.

L378-380 I think some subtle changes here would make this clearer: "These percentiles account for the uncertainty of the CMIP models weighted to account for the likelihood of their ECS, and for the uncertainty of the threshold on liquid water production in excess when we investigate the potential for ice shelf hydrofracturing."

⇒ This has been modified as suggested.

Fig 8 caption missing reference ("section??")

⇒ This has been corrected.

L484 Earlier you use the spelling Dröning Maud Land – now it's Dronning Maud Land. Please use one for consistency.

⇒ This has been corrected to "Dronning".

Discussion – a brief discussion of MAR's limitations (e.g. lack of ponding/routing, no dynamic ice sheet etc.), as well as how well the method may be applied to other RCMs would be interesting to include.

⇒ We have added the following sentences to the Discussion:

*The absence of physical representation of ponding and horizontal routing of liquid water nonetheless remains a major caveat of all current RCMs. Another important limitation of RCMs is the use of a constant ice sheet elevation, although the melt–elevation feedback seems to become important only after 2200 in the SSP5-8.5 scenario (Coulon et al., 2023).*

L518 "Other RCMs, possibly combined to elaborated firn models" – do you mean "combined with elaborated firn models"?

⇒ Yes, this has been corrected.

L554 "built" ◊ "build"

⇒ This has been corrected.

L558 "models" ◊ "models'" or "model"

⇒ This has been corrected.

Figs C1-C3 I'm impressed by the similarities between the directly simulated and emulated results. The only major difference I can really pick out is a different sign between MAR/emulation over the Ross ice shelf, which is evident in all three figures. Can you comment on this?

⇒ We have added the following comment to Appendix D:

*For a reason that remains elusive, the largest mismatch is found on the Ross ice shelf, where the emulation produces a negative SMB around 2100 for the three applications, while it remains mostly positive in the original MAR simulation.*

L619-621 Missing figure references

⇒ This has been corrected.

L620 Think this should be "In East Antarctica, a runoff anomaly first prevails"

⇒ This has been corrected to "In East Antarctica, runoff anomalies first prevail"

Fig D1 caption missing section reference

⇒ This has been corrected.